
# Global scale benefit-cost analysis of coastal flood adaptation to different flood risk drivers

Timothy Tiggeloven[1], Hans de Moel[1], Hessel C. Winsemius[2,5], Dirk Eilander[1,2], Gilles Erkens[2], Eskedar Gebremedhin[2], Andres Diaz Loaiza[1,6], Samantha Kuzma[4], Tianyi Luo[4], Charles Iceland[4], Arno Bouwman[3], Jolien van Huijstee[3], Willem Ligtvoet[3], Philip J. Ward[1]

[1]Institute for Environmental Studies (IVM), Vrije Universiteit Amsterdam; The Netherlands
[2]Deltares, Delft, The Netherlands
[3]PBL Netherlands Environmental Assessment Agency; The Hague, The Netherlands
[4]World Resources Institute; Washington DC, USA
[5]Water Management Department, Delft University of Technology, Delft, The Netherlands
[6]Hydraulic Structures and Flood Risk, Delft University of Technology, Delft, The Netherlands

*Correspondence to*: Timothy Tiggeloven (timothy.tiggeloven@vu.nl)

**Abstract.** Coastal flood hazard and exposure are expected to increase over the course of the 21st century, leading to increased coastal flood risk. In order to limit the increase in future risk, or even reduce coastal flood risk, adaptation is necessary. Here, we present a framework to evaluate the future benefits and costs of structural protection measures at the global scale, which accounts for the influence of different flood risk drivers (namely: sea-level rise, subsidence, and socioeconomic change). Globally, we find that the estimated expected annual damage (EAD) increases by a factor of 150 between 2010 and 2080, if we assume that no adaptation takes place. We find that 15 countries account for approximately 90% of this increase. We then explore four different adaptation objectives and find that they all show high potential to cost-effectively reduce (future) coastal flood risk at the global scale. Attributing the total costs for optimal protection standards, we find that sea-level rise contributes the most to the total costs of adaptation. However, the other drivers also play an important role. The results of this study can be used to highlight potential savings through adaptation at the global scale.

## 1 Introduction

In recent years, the effects of climate change on coastal flood hazards and its impacts on society have been studied extensively. The Fifth Assessment Report (AR5) of the Intergovernmental Panel on Climate Change (IPCC) reports that it is likely that we will face a mean sea-level rise by the end of the 21st century in the range of approximately 0.3 – 1 meter compared to compared to 1986-2005 and that impacts on society will be vast (IPCC, 2014). According to a recent study by Raftery et al. (2017)., it is unlikely that the Paris agreement's aim of keeping global warming below a 2°C increase by the end of the 21st century will be met. This may lead to an increase in storm surges (Tebaldi et al., 2012) and extreme sea levels (Vousdoukas et al., 2017). Together, these increases in sea-level and a possible change in storminess will lead to increased flood hazards, as well as threats to shorelines, wetlands, and coastal development (Ericson et al. 2006; Hinkel et al., 2013). Moreover, flood hazard is expected to increase as a result of subsidence. In many deltas and estuaries, groundwater extraction is a major factor





contributing to this subsidence (Hallegatte et al., 2013). During the 20[th] century, the coasts of Tokyo, Shanghai and Bangkok subsided by several meters (Nicholls et al., 2008) and subsidence is expected to continue to affect coastal flood risk in the

future (Dixon et al., 2006). Global coastal flood risk is also expected to increase in the future as a result of increasing exposure, due to growth in population and wealth, and economic activities in flood-prone areas (Güneralp et al., 2015; Jongman et al., 2012; Neumann et al., 2015; Pycroft et al., 2016).

Today, on average 10% of the world population and 13% of the total urban area in low elevation coastal zones is located less than 10 meters above sea level (McGranahan et al., 2007). In addition, 1.3% of global population is estimated to be exposed

to a 1 in 100-year flood (Muis et al., 2016). In the coming century, these people and areas are projected to face increases in coastal flood risk (Brown et al., 2018; Hallegatte et al., 2013; Hinkel et al., 2014; Jongman et al., 2012; Merkens et al., 2018; Neumann et al., 2015).

In order to prevent this increase in coastal flood risk, or even to reduce risk below today's levels, adaptation measures are necessary. The importance of climate change adaptation and disaster risk reduction is recognized in several global agreements,

such as the Paris Agreement (United Nations Framework Convention on Climate Change, 2015) and the Sendai Framework for Disaster Risk Reduction (United Nations Office for Disaster Risk Reduction, 2015). The Sendai Framework sets specific targets for reducing risk by 2030, such as reducing the direct disaster economic loss in relation to GDP and substantially reducing the number of affected people globally.

Recent studies have shown that adaptation measures hold a large potential for significantly reducing this future flood risk

(Diaz, 2016; Hinkel et al., 2014; Lincke and Hinkel, 2018). However, the number of global scale studies in which the benefits and costs of disaster risk reduction and adaptation are explicitly and spatially accounted for remains limited. Existing studies have assessed the effect of climate change and/or socioeconomic change (Hallegatte et al., 2013; Hinkel et al., 2014; Vousdoukas et al., 2016). Lincke & Hinkel (2018) assessed the cost-effectiveness of structural protection measures against sea-level rise and population growth using the DIVA model. They found that structural adaptation measures are for 13% of

the global coastline feasible to invest in.

In this paper, we develop a model to evaluate the future benefits and costs of structural adaptation measures at the global scale. We use it to extend the current knowledge on the cost-effectiveness of structural adaptation measures in several ways. Firstly, we include subsidence due to groundwater extraction. Secondly, we assess the benefits and costs of several adaptation objectives. Thirdly, we attribute the costs of adaptation to different drivers (namely sea-level rise, subsidence and change in

exposure).

## 2 Methods

The overall methodological framework is summarized in Figure 1, and consists of the following main steps: (1) flood risk estimation; (2) adaptation costs estimation; (3) benefit-cost analysis for four adaptation objectives; and (4) attribution of the total costs to the different drivers. Each of these steps is described in detail in the following subsections. In brief, flood risk is

estimated as a function of hazard, exposure and vulnerability (United Nations Office for Disaster Risk Reduction, 2016). In





the risk model, Expected Annual Damage (EAD) is calculated for different scenarios with and without adaptation, with the difference between these two representing the benefits. The costs are calculated by estimating the dimensions of the required dikes (height and length) and multiplying these by their unit costs. Maintenance costs are also included in the cost model. A benefit-cost analysis is performed for four adaptation objectives, and finally the costs of adaptation are attributed to several

risk drivers. The methodological steps takes are explained in detail in Ward et al. (2019), on which the following descriptions are based.

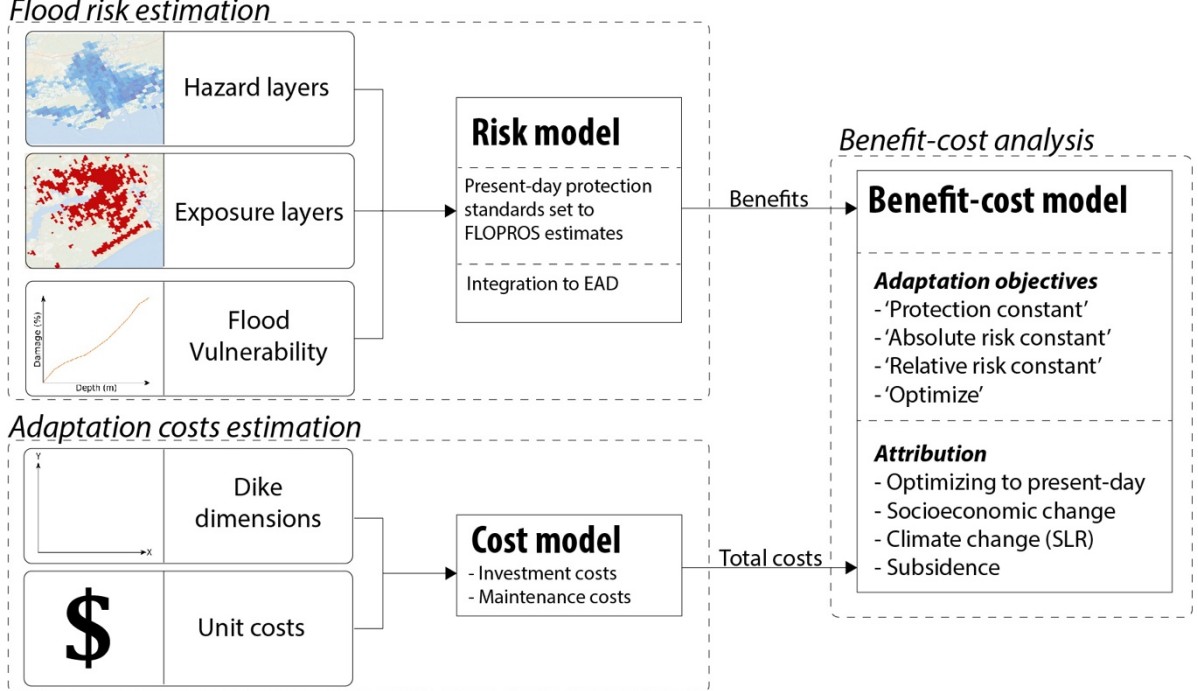

**Figure 1: Overview of models and data layers for assessing flood risk, costs of adaptation and attribution of different drivers.**

## 2.1 Flood risk estimation

We use hydrodynamic simulations of tide and surge, and scenarios of regional sea level rise, as input to a coastal inundation model, in order to generate hazard maps for several return periods (2, 5, 10, 25, 50, 100, 250, 500 and 1000 years). These are combined with exposure maps and vulnerability curves (depth-damage functions) in the impact assessment model, using a setup similar to the GLOFRIS impacts module developed by Ward et al. (2013) and extended for future simulations by Winsemius et al. (2016). The global coastal flood impacts are assessed at a horizontal resolution of $30'' \times 30''$ and simulated

for the different return periods. After calculating the impacts for the different return periods, EAD is calculated by taking the integral of the exceedance probability-impact curve (Meyer et al., 2009). Figure 2 shows the different input layers for the flood risk assessment and benefit-cost analyses (note that different sea-level rise and socioeconomic scenarios are used, and just one is shown in Figure 2 as example). The following section describes the flood risk simulations in detail.

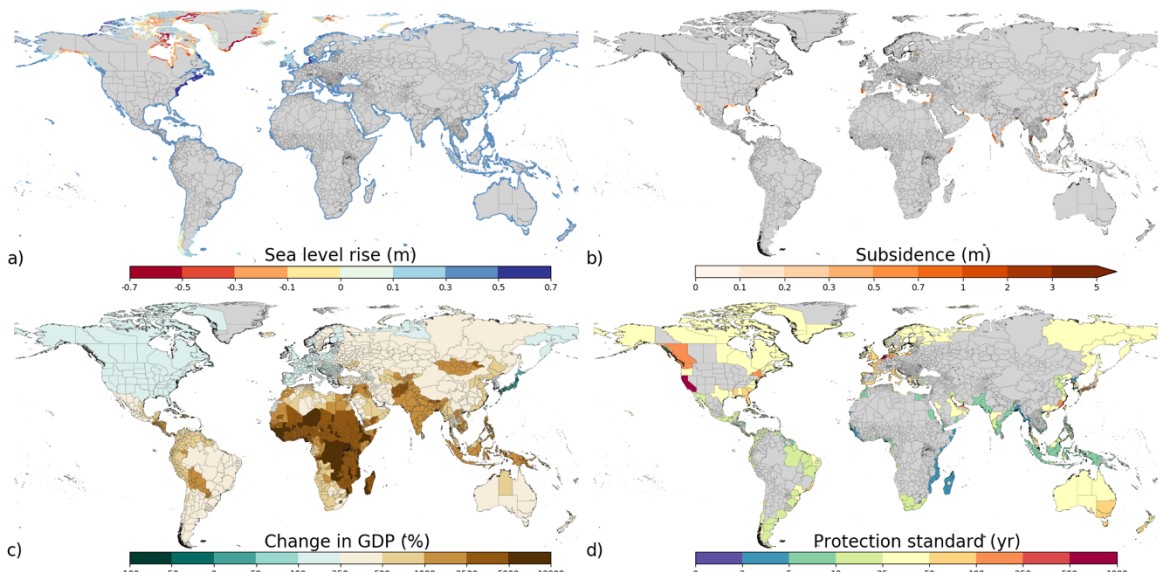

**Figure 2: Input layers for the benefit-cost analyses: (a) sea level rise for the RCP4.5 scenario in 2080; (b) subsidence in 2080; (c) Change in GDP for the SSP2 scenario in 2080; and (d) current protection standards estimated with the FLOPROS modelling approach.**

### 2.1.1 Flood hazard

**Current flood hazard.** In order to simulate coastal inundation hazard, we use extreme sea levels from the Global Tide and Surge Reanalysis (GTSR) dataset by Muis et al. (2016) as input to an inundation model. GTSR has been shown to perform well (Muis et al., 2017) for extratropical regions and contains a database of extreme water levels for different return periods, based on the Global Tide and Surge Model (GTSM). Surge is simulated using wind and pressure fields from the ERA-Interim reanalysis (Dee et al., 2011), and tide is simulated using the Finite Element Solution 2012 (FES2012) model (Carrère and Lyard, 2003). In this modelling scheme, wind (or surface) waves are not included. As tropical cyclones are poorly represented in the input climate dataset, we use a version of GTSR enriched using a historical storm track archive to represent tropical cyclones. These tropical cyclones were simulated using the IBTrACS (International Best Track Archive for Climate Stewardship) archive, which provides a dataset of historical best tracks.

To calculate overland inundation from near-shore tide and surge levels we used a GIS-based inundation routine, similar to Vafeidis et al. (2019). Extreme sea levels from the nearest GTSR location are projected at the coastline. Then, inundation takes place in areas that are hydraulically connected to the sea for that extreme sea level. The model uses the Multi-Error-Removed Improved-Terrain (MERIT) DEM (Yamazaki et al., 2017) at a 30" x 30" resolution as underlying topography. We accommodate three important factors in the inundation routine that are not regularly taken into account in global scale coastal inundation modelling:

- We use a resistance factor to simulate the reduction of flooding land inwards as tides and storm surges have a limited time span. We apply this factor over a Euclidean distance from the nearest coast line point. The resistance factor was





set to 0.5m/km. Haer et al. (2018) showed the maps to perform well against past flood events in their study in Mexico. Several other studies also use attenuation factors varying between 0.1 and 1.0 m/km (Vafeidis et al., 2019).

- We multiply the resistance factor by a weight, proportional to the amount of permanent water in each cell within the Euclidean pathway towards a land cell under consideration. In this way, grid cells that are marked as land within the terrain model, but in fact represent areas with large amounts of open water are correctly simulated as cells with low resistance. We estimate fractions of permanent water using a 30-year monthly surface water mask dataset at 30 meter resolution, derived from LandSAT archive (Pekel et al., 2016).

- We apply a spatially varying offset between Mean Sea Level according to the FES2012 model, and the datum used by the terrain model MERIT (EGM96) to ensure that the zero datum of our terrain and our extreme sea levels from GTSR are the same.

**Future flood hazard.** For future hazard simulations we use sea level changes, to simulate future extreme sea levels, and subsidence estimates to estimate how the terrain may change. Gridded regional sea-level rise projections for RCP4.5 and RCP8.5 are obtained from the RISES-AM project (Jevrejeva et al., 2014). The sea-level rise from this project is simulated as a range of probabilistic outcomes. For this paper the 50th percentile is used, and to assess the sensitivity of the results we also use the 5th and 95th percentiles as input for the inundation model. We include sea-level rise in the inundation routine by adding this additional water level to the extreme sea level. Sea-level rise in 2080 for the RCP4.5 scenario and 50th percentile is shown in Figure 2a. In this simulation, most of the regions will face a sea-level rise between the 0.3 and 0.5 meters. Close to the poles, sea level may decrease due to a decline in gravitational forces of the melting ice caps.

Subsidence rates are taken from the SUB-CR model by Kooi et al. (2018), which models subsidence using three existing models, namely the hydrological model PCR-GLOBWB integrated with the global MODFLOW groundwater model (de Graaf et al., 2017; Sutanudjaja et al., 2018), and a land subsidence model (Erkens and Sutanudjaja, 2015), focussing on groundwater levels and resulting subsidence. In this approach, subsidence is modelled due to groundwater extraction, which is the dominant factor of human-induced subsidence in many coastal areas (Erkens et al., 2015; Galloway et al., 2016). The effects of subsidence, simulated at the resolution of 5' x 5' and spatially interpolated to 30' x 30' resolution, are included in the inundation model by adding the subsidence estimates to the MERIT terrain. Subsidence in 2080 is shown in Figure 2b and reaches up to 5-7m in regions in China. Unlike sea level rise, subsidence does not take place along every coastline and is instead projected as a regional phenomenon.

### 2.1.2 Flood exposure

In our modelling scheme, exposure is represented by maps of built-up area and estimates of maximum damage for three different land use classes in built-up areas. The GLOFRIS model uses current and future built-up area, current and future GDP, and maximum damages on the country level as input. The FLOPROS modelling approach (see section 2.1.5) has current data on built-up area, population and GDP as input. In the following sections, we describe the exposure data for the current and future simulations.



**Current exposure.** Current built-up area with a resolution of 5' x 5' are taken from the HYDE database (Klein Goldewijk et al., 2010) and later regridded to the 30" x 30" resolution. Built-up area refers to all kinds of built-up areas and artificial surfaces. Current maximum economic damages are estimated using the methodology of (Huizinga et al., 2017). To convert construction costs to maximum damages, several adjustments are carried out using the suggested factors by Huizinga et al. (2017). As a proxy for an approximation of percentage area per occupancy type, we set the urban grid cells of the layers from the HYDE database to 75% residential, 15% commercial and 10% industrial, based on a study by (BPIE, 2011) and a comparison of European cities' share of occupancy type of the CORINE Land Cover data (EEA, 2016). Following Huizinga et al. (2017), the density of buildings per occupancy types are set to 20% for residential and 30% for commercial/industrial.

In order to normalize current risk we use GDP per capita taken from the SSP database of IIASA, distributed spatially according to the ORNL LandScan 2010 population count map (Bright et al., 2011). As the total population per country in this map is different to the 2010 population stated in the SSP database, we use a correction factor per country to adjust the population per cell.

**Future exposure.** Future simulations of built-up area are taken from Winsemius et al. (2016) at a resolution of 30" x 30". Using the method described by Jongman et al. (2012), these simulations were computed using changes in gridded population and urban population for different SSPs derived from the GISMO/IMAGE model (Bouwman et al., 2006).

To estimate future maximum damages, we scale the current values with the GDP per capita per country from the SSP database. Boundaries of countries are derived from the Global Administrative areas dataset (GADM, 2012). In order to calculate future risk relative to GDP, future GDP values are taken from Van Huijstee et al. (2018).

### 2.1.3 Flood vulnerability

Vulnerability to flood depth of urban areas is estimated by using different global flood depth-damage functions for each occupancy type and are taken from Huizinga et al. (2017). The resulting damages are represented as percentage of the maximum damage, reaching maximum damages at a water level depth of 6 meters.

### 2.1.4 Integration to EAD

With the urban damages, calculated for the different return periods, risk is computed and expressed in terms of expected annual damages (EAD). We use a commonly used method in risk assessment to calculate EAD by taking the integral of the exceedance probability-impact (risk) curve (Meyer et al., 2009) and can be written as

$$EAD = \int_{p=0}^{1} D_\theta(p)dp, \tag{1}$$

where $EAD$ is 'risk' per year, $D$ is the urban damage (or impact), $\theta$ the vulnerability, and $p$ denotes the annual probability of non-exceedance (protection standard divided into 1). To fit a protection standard of a coastal region in the risk computation, the risk curve is truncated at the exceedance probability of the protection standard (expressed as a return period). To estimate the definite integral, we use the trapezoidal approximation. As data on protection standards of coastal regions are not available





for many regions, we estimate current protection standards for coastal regions using the FLOPROS modelling approach (Scussolini et al., 2016), as is described in section 2.1.5.

### 2.1.5 FLOPROS modelling approach

In order to assess the benefits and costs of adaptation objectives, information on current protection standards is needed. We use the FLOPROS modelling approach (Scussolini et al., 2016) to estimate these protection standards. This section contains a

brief description of the coastal protection standards estimated with the FLOPROS modelling approach.

Using current exposure data and EAD data from the GLOFRIS model as input, the current protection standards are estimated using the FLOPROS modelling approach. Figure 2d shows the estimated FLOPROS flood protection standards for each coastal sub-national unit. Higher protection standards can be found at regions with high economic activity and high asset exposure. Regions with low risk have lower estimated protection standards. Regions without modelled risk in the GLOFRIS model are

excluded. This occurs in regions where we have no data on exposure or no coastal inundation is simulated. These protection standards are used in our paper as the current protection, on top of which the future costs of dike heightening are calculated. The protection standards for The Netherlands are manually set to 1000. This is because, for whole of The Netherlands protection standards are known to be higher than 1000.

### 2.1.6 Estimating the benefits of adaptation

In order to calculate the benefits of adaptation, EAD is calculated for every year of the lifetime of the dike for a certain return period and subtracted from the EAD for every year without adaptation. The lifetime of the dike is set to expire in 2100 and the building period is set to 20 years. During this period EAD is assumed to increase linearly. The results are summed to get the total benefits of adaptation.

### 2.2 Cost estimation

To estimate the costs associated with the different adaptation objectives, we use the same methodology as Ward et al. (2017), which calculates the costs of flood protection by summing the maintenance and investment costs over time for raising dikes to prevent flooding. The following section describes the calculation of costs of adaptation and the adaptation objectives in more detail.

In order to calculate the costs of adaptation, first dike heights need to be calculated. The current dike height calculations are

taken from a recent study by van Zelst et al. (2019). Their methodology is to first derive coastal segments and perpendicular coast-normal transects (766,034 transects in total). For each transect, bed levels are constructed and subsequently, hydrodynamic conditions and wave attenuation are derived. Lastly, the resulting sea water levels are translated into dike heights. The coastlines are derived from OpenStreetMap (OSM) and moved 100 m land inwards to smoothen the coast lines and to position the lines at a likely place to establish a dike system. Transects are derived perpendicular to the coastlines for

each 1" x 1"-cell that has a coastline segment. Each transect is described by its slope, ocean bathymetry, foreshore, elevation and surge levels among other things. To capture most foreshores, the transects are stretched 4 km land inward and seaward.


The main source of bed level data is the Earth Observation (USGS Landsat and Copernicus Sentinel 2) based high resolution intertidal elevation map (20 m horizontal and 30-50 cm vertical accuracy) of Calero et al. (2017). As this dataset does not contain data for all bed levels along the transects, the gaps are filled by ocean bathymetry data from GEBCO (30'', 10 m
vertically) and topography data from MERIT (3'', 2 m vertically). The water levels are derived from the GTSR dataset (Muis et al., 2016) and corresponding wave conditions at different return periods from the ERA-Interim reanalysis (Dee et al., 2011). With a lookup-table, consisting of numerical modelling results, the wave attenuation over the foreshore is determined. Due to the unknown direction, incoming waves are assumed to run perpendicular to the coast. Finally, current dike heights in respect to the surge level are calculated with the empirical EuroTop formulations (Pullen et al., 2007) and are based on a standard 1:3
dike profile without berms and with a maximum allowed overtopping discharge of 1 L/m$^2$/s. This is representative for a low-cost dike. We exclude coastlines where there is no built-up area or no inundation is simulated.

In order to calculate future dike heights, sea-level rise from the RISES-AM project (Jevrejeva et al., 2014), is used in the calculation of the crest heights for different return periods. This is done by adding sea level rise directly to the crest height. Next to sea level rise, future dike heights are calculated with subsidence levels (see section 2.1.1.). Subsidence is assumed to
take place directly on the dike and therefore computed on the crest height, which is similar for sea level rise calculations.

The costs of raising dikes are estimated by calculating the total length of dike heightening per grid cell and multiplying by a unit cost set to USD 7 million km·m based on reported costs in New Orleans (Bos, 2008). This value of US$ 7 million km·m is within a reasonable range when compared to various studies (Aerts et al., 2013; Jonkman et al., 2013; Lenk et al., 2017). This includes investment cost, groundwork-, construction- and engineering costs, property or land acquisition, environmental
compensation, and project management. Subsequently, the costs are converted to US$2005 Power Purchasing Parity (PPP) using GDP deflators from the World Bank and average annual market exchange rates from the European Central Bank for each country. Construction index multipliers, based on civil engineering construction costs, adjust the construction costs to account for differences between countries (Ward et al., 2010). The lengths of the dikes are estimated using the 766,034 coastline transects. Maintenance costs are represented as percentages of investment costs and are set to 1% per year.

**2.3 Benefit-cost analysis**

Finally, a benefit-cost analysis is performed by calculating the benefits and costs for adaptation until 2100 for sub-national regions. These regions are defined as the next administrative unit below national scale in the Global Administrative Areas Database (GADM). The benefits and costs are discounted with a discount rate of 5% until 2100 (lifespan of investment) and with Operation and Maintenance (O&M) costs of 1%. It is assumed that investments are made in 2020 and construction is
finished in 2050. During this time period, benefits and costs for investment are assumed to increase linearly. We use Net Present Value (NPV) shown in Eq. (2) and Benefit-Cost Ratios (BCR) shown in Eq. (3) as indicators of economic efficiency.

$$NPV = \sum_{t=1}^{n} \frac{B_t - C_t}{(1+r)^t} - C_0 \qquad (2)$$


$$BCR = \left.\frac{\sum_{t=1}^{n}\frac{B_t}{(1+r)^t}}{}\middle/\left(\sum_{t=1}^{n}\frac{C_t}{(1+r)^t} + C_0\right),\right. \tag{3}$$

where $t$ denotes the time in years, $n$ the lifespan of the investment, $r$ the discount rate, $B_t$ the benefits per year, $C_t$ costs per

year expressed as maintenance costs, and $C_0$ the initial investment costs.

### 2.3.1 Adaptation objectives

For the benefit-cost analysis, four future investment objectives are explored: (1) 'Protection constant', which keeps protection levels in the future the same as current protection levels; (2) 'Absolute risk constant', which calculates future protection standards when the absolute value for EAD is kept the same as current; (3) 'Relative risk constant', which calculates future

protection standards when EAD as a percentage of GDP is kept the same as current; and (4) 'Optimize', which calculates future protection standards by maximizing NPV. The future protection standards for the four adaptation objectives are estimated at discrete intervals (2, 5, 10, 25, 50, 100, 250, 500 and 1000 years). The future protection standards when no adaptation takes place are calculated by assuming that dikes are maintained at the current height, but with no additional heightening. In the 'optimize' adaptation objective, only regions with BCR greater than 1 are included; no adaptation takes

place for regions with BCR less than 1.

### 2.3.2 Attribution of costs

In order to attribute costs to different drivers, the following method is used. For the 'optimize' adaptation objective, the costs are attributed to four terms: (1) optimisation under current conditions (CUR); (2) socioeconomic change (SEC); (3) sea level rise driven by climate change (SLR); and (4) subsidence driven by groundwater depletion (SUB). The following conceptual

equations illustrate the attribution methodology:

$$A_{CUR} = NPV_{CUR}/NPV_{ALL}, \tag{4}$$

$$A_{SEC} = (NPV_{SEC} - NPV_{CUR})/NPV_{ALL}, \tag{5}$$

$$A_{SLR} = NPV_{SLR\ (baseline\ protection\ SEC)}/NPV_{ALL}, \tag{6}$$

$$A_{SUB} = NPV_{SUB\ (baseline\ protection\ SEC)}/NPV_{ALL} \tag{7}$$

Equation (4-7) show the attribution calculation with $A$ the attribution and $NPV$ the net present value calculated with Eq. (4) The subscripts denote the attribution terms: $CUR$ refers to optimizing in current conditions; $SEC$ refers to socioeconomic change; $SLR$ refers to sea-level rise; and $SUB$ refers to subsidence. $ALL$ refers to when all risk drivers are taken into account. In the subscript between brackets, the baseline protection standard used during the calculation of NPV is indicated. Because the 'optimize' adaptation objective is an optimisation and not all regions have optimised their protection standards for the

current climate, this last term must be accounted for. The optimisation term is the costs of maximizing NPV with current conditions ($NPV_{CUR}$). Subsequently, the costs for socioeconomic change are computed by taking the difference in costs between $NPV_{CUR}$ and maximizing NPV when only socioeconomic change is taken into account ($NPV_{SEC}$). To determine the





attribution of costs for climate change, the baseline protection is set to the protection standards associated with the $NPV_{SEC}$ term. Subsequently, the costs are estimated by maximizing NPV when both sea-level rise and socioeconomic change are taken

into account ($NPV_{SLR}$). The attribution of subsidence is the same procedure as with $NPV_{SLR}$, by swapping the sea-level rise driver with the subsidence driver ($NPV_{SUB}$). All attributions of costs are expressed in percentages with reference to maximizing NPV for future conditions ($NPV_{ALL}$), which is the same as the 'optimize' adaptation objective.

In some cases, the percentages of the different drivers do not add up to 100%. This is the case when absolute dike heights associated with $NPV_{SEC}$ are higher than $NPV_{ALL}$ (in other words: adding climate change and subsidence would actually result

in lower optimal dike heights in the benefit-cost analysis). In these cases, we set attribution for $ATR_{SEC}$ to 100%, and $ATR_{SLR}$ and $ATR_{SUB}$ to 0%. Another exception is when optimal protection standards for $NPV_{SEC}$ are higher than $NPV_{SLR}$ or $NPV_{SUB}$. This occurs when the increase in absolute dike height in the optimization is lower than the effect of sea-level rise or subsidence, and results in a lower protection standard. For all other cases, except the two mentioned above, the sum adds to 100%.

## 3 Results and Discussion

In this section, we first present an assessment of current and future risk without adaptation. Next, we present global benefit-cost analyses for the different adaptation objectives. Then, we present the results of the benefit-cost analyses and the attribution of costs to different drivers at the regional scale. Finally, we assess the sensitivity of the results to changes in various parameters.

### 3.1 Overview of future flood risk assuming no adaptation.

Globally, the estimated EAD increases by a factor of 150 between 2010 and 2080, if we assume that no adaptation takes place. Figure 3 shows the top 15 countries that contribute to this coastal flood risk, in 2010 (Figure 3a) and 2080 (Figure 3b) – note the different scales on the x-axis. China, Bangladesh, and India have the highest flood risk in absolute terms in 2010. In 2080, these three countries remain in the top four if no adaptation takes place, and are joined by the Netherlands. The 15 countries shown account for 89% of coastal flood risk worldwide in 2010 (US$19.6 billion per year globally). Although the countries in

the top 15 change between current and future assuming no adaptation, the total share of EAD residing in the top 15 countries remains approximately the same: 87% of global flood risk in 2080 if no adaptation takes place (US$3 trillion per year globally for RCP4.5/SSP2 and US$6.8 trillion for RCP8.5/SSP5).





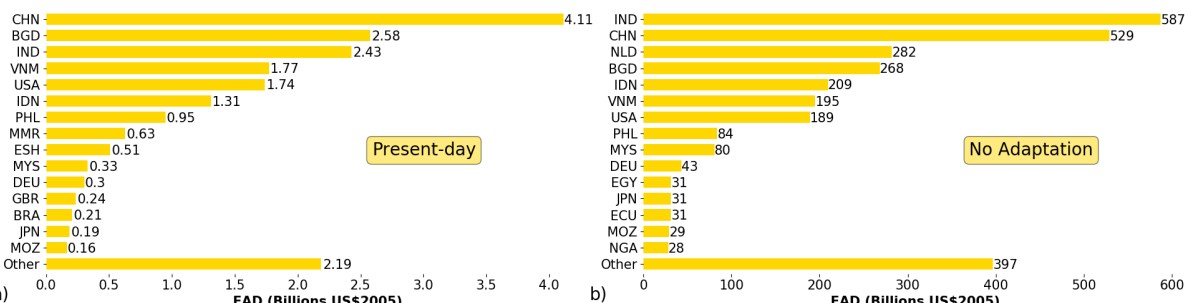

**Figure 3: Top 15 countries with coastal flood risk in (a) current conditions; and (b) 2080 if no adaptation takes place for the scenario RCP4.5/SSP2. Note that the countries and value on the x-axis change for each graph.**

**3.2 Global scale assessment of flood risk under the different adaptation objectives.**

The flood damage simulations were carried out for two different sea-level rise scenarios (RCPs) and five different socioeconomic scenarios (SSPs). All the results are shown for two scenarios, namely RCP4.5/SSP2 and RCP8.5/SSP5. The former is used for a 'middle of the road' scenario with medium challenges and adaptation (Riahi et al., 2017) that can broadly be aligned with the Paris agreement targets (Tribett et al., 2017), while the latter is used as a 'fossil-fuel development' world (Kriegler et al., 2017). Results of the other combinations can be found in the supplementary data.

For all four adaptation objectives, a globally aggregated overview of the benefits, costs, BCR, and NPV is provided in Table 1. All objectives have a positive NPV and BCR higher than 1, indicating that globally the benefits in terms of reduced risk would exceed the investment and maintenance costs. Note that only regions with positive NPV are included for the 'optimize' adaptation objective. The 'absolute risk constant' adaptation objective has the lowest BCR, while the 'optimize' adaptation objective has, by definition, the highest BCR. Higher costs and benefits are found for the RCP8.5/SSP5 scenario compared to the RCP4.5/SSP2 scenario, as a result of the larger EAD (and therefore avoided EAD) under this scenario. On average, the costs are ca. 25% larger in the former, and the benefits roughly double.

**Table 1: Global overview of benefit-cost analysis for the different adaptation objectives (benefits, costs, and NPV are in billion US$2005).**

|  |  | Benefits | Costs | BCR | NPV |
|---|---|---|---|---|---|
| 'Protection constant' | RCP4.5/SSP2 | 9,705 | 144 | 67 | 9,561 |
|  | RCP8.5/SSP5 | 18,729 | 176 | 106 | 18,552 |
| 'Absolute risk constant' | RCP4.5/SSP2 | 11,550 | 307 | 38 | 11,243 |
|  | RCP8.5/SSP5 | 23,020 | 399 | 58 | 22,620 |
| 'Relative risk constant' | RCP4.5/SSP2 | 11,027 | 186 | 59 | 10,840 |
|  | RCP8.5/SSP5 | 22,101 | 224 | 99 | 21,878 |
| 'Optimize' | RCP4.5/SSP2 | 11,550 | 152 | 76 | 11,398 |
|  | RCP8.5/SSP5 | 23,031 | 208 | 111 | 22,823 |



The top 15 countries that contribute the most to coastal flood risk for the four adaptation objectives for RCP4.5/SSP2 in 2080 are shows in Figure 4. The total share of EAD residing in the top 15 countries remains approximately the same: 94% of global flood risk in the 'protection constant' adaptation objective (US$ 767 billion per year globally); 93% in the 'absolute risk constant' adaptation objective (US$238 billion per year); 90% in the 'relative risk constant' adaptation objective (US$421

billion per year); and 91% in the 'optimize' adaptation objective (US$242 billion per year globally). Note that EAD can increase in the future for the 'absolute risk constant' adaptation objective in certain regions as we cap protection standards at 1000. The simulated optimal protection standards of the Netherlands are lower than in the 'protection constant' adaptation objective, resulting in a high future EAD of US$60.9 billion per year. This is because the simulated marginal costs of dike heightening up to a protection standard of 1000 years outweigh the marginal benefits. However, it should be noted that the

benefits do exceed the costs up to a 1000-year protection standard, and that if this were implemented, the future EAD for the Netherlands in the 'optimize' adaptation objective would therefore be much lower than shown in Figure 4. Supplementary Figure S1 shows the top 15 countries for RCP8.5/SSP5.

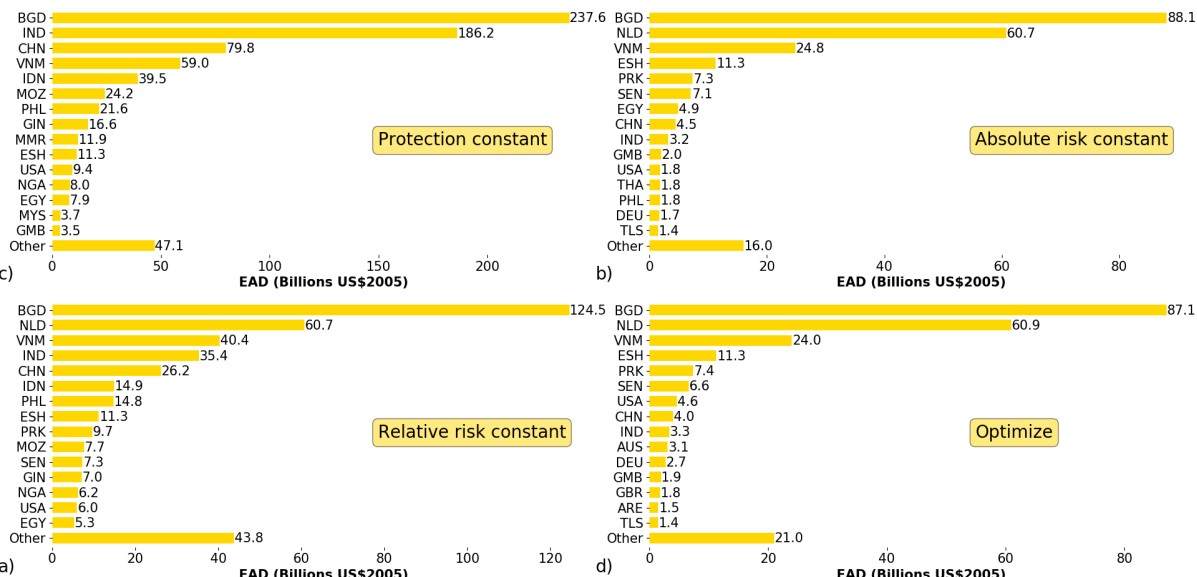

**Figure 4: Top 15 countries with coastal flood risk in (a) 2080 if protection standards are kept constant; (b) 2080 if absolute risk is**
**kept constant; (c) 2080 if relative risk is kept constant; and (d) 2080 if protection standards are optimized for the scenario**
**RCP4.5/SSP2. Note that the countries and value on the x-axis change for each graph.**

### 3.3 Regional scale assessment of flood risk under the different adaptation objectives.

In order to show spatial patterns of the four adaptation objectives, the following results are shown at the sub-national scale in Figures 5-8. Here, results are shown for RCP4.5/SSP2 only. The same results for RCP8.5/SSP5 can be found in Supplementary

Figures S2-S5, and the data for all scenario combinations can be found in Supplementary Data. Although there are some differences between the results for RCP4.5/SSP2 and RCP8.5/SSP5, the overall patterns are very similar.

In the **'protection constant'** adaptation objective, the benefits outweigh the costs for the majority of the regions (82%; 643 of the 784 sub-national regions assessed). Nevertheless, this would still lead to an increase in relative risk (i.e. EAD as a



percentage of GDP) in the future for 82% (641) of the regions assessed. Therefore, only raising dikes to keep up with the
current protection standard would lead to a substantial increase in future risk in the majority of the world's regions for scenario
RCP4.5/SSP2. Sub-national regions in South Asia, Southeast Asia, eastern Australia, the east and west coast of North America,
and parts of Europe have the highest BCR and NPV (Figure 5). Note that the protection standards (Figure 5a) are the same as
the current protection standards (Figure 2d).

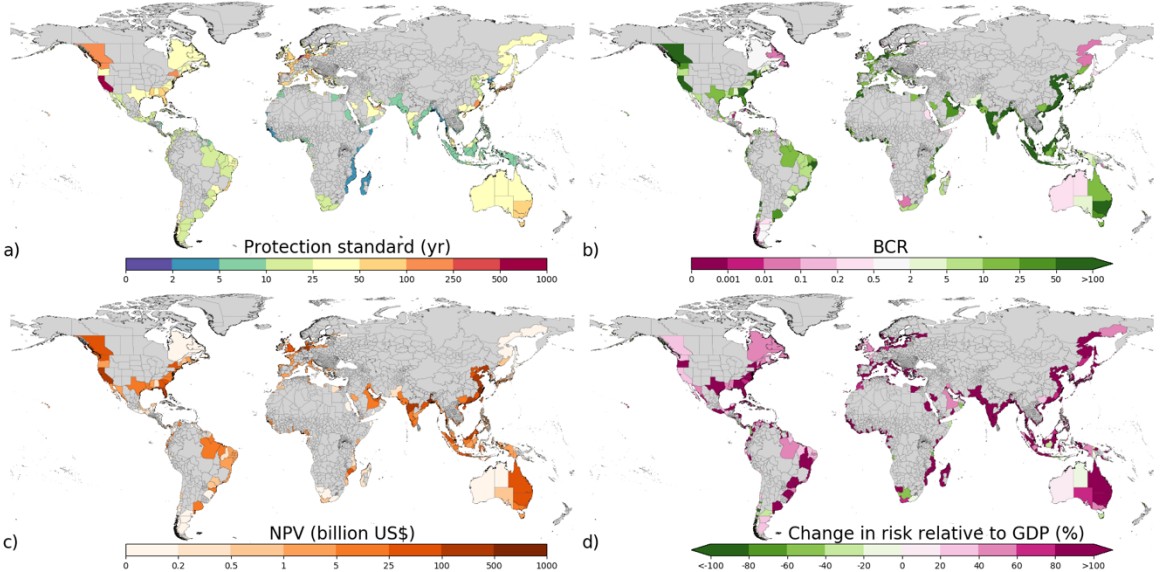

**Figure 5: 'Protection constant' adaptation objective results of (a) protection standards; (b) BCRs; (c) total NPV; and (d) change in risk relative to GDP for RCP4.5/SSP2. Note that the protection standards (a) are the same as FLOPROS estimates.**

In the **'absolute risk constant'** adaptation objective (Figure 6), it is clear that dikes would need to be upgraded to have high
protection standards (usually between 100 and 1000 years) in order to keep risk constant at current levels. The costs to achieve
this are high (globally, more than twice as high as under the 'protection constant' adaptation objective) and therefore a lower
number of sub-national regions (79%; 623) have a positive BCR, although this is still very high. In most sub-national regions,
the risk relative to GDP decreases in the future if this adaptation objective is implemented, although 5% (38) of the sub-national
regions show an increase in risk relative to GDP.

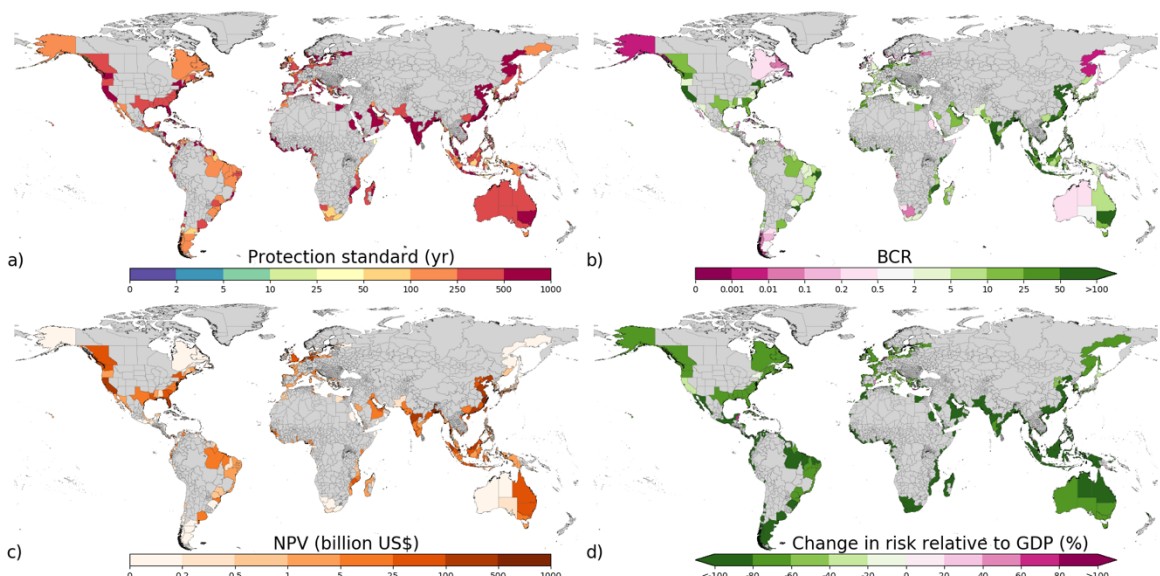

**Figure 6: 'Absolute risk constant' adaptation objective results of (a) protection standards; (b) BCRs; (c) total NPV; and (d) change**
**in risk relative to GDP for RCP4.5/SSP2.**

In the **'relative risk constant'** adaptation objective (Figure 7), the protection standards required are generally lower than in
the 'absolute risk constant' adaptation objective. The highest protection standards required are found in East Asia and parts of
North America. A similar number of sub-national regions have a BCR higher than 1 as is the case for 'absolute risk constant',
namely 79% of the sub-national regions assessed. To keep relative risk constant or absolute risk constant some sub-national
regions need to have a future protection standard that is higher than 1000-year (the highest return period assessed in this study).
Because of this, the relative change in risk in the 'Relative risk constant' adaptation objective increases for 5% (36) of the
regions assessed.


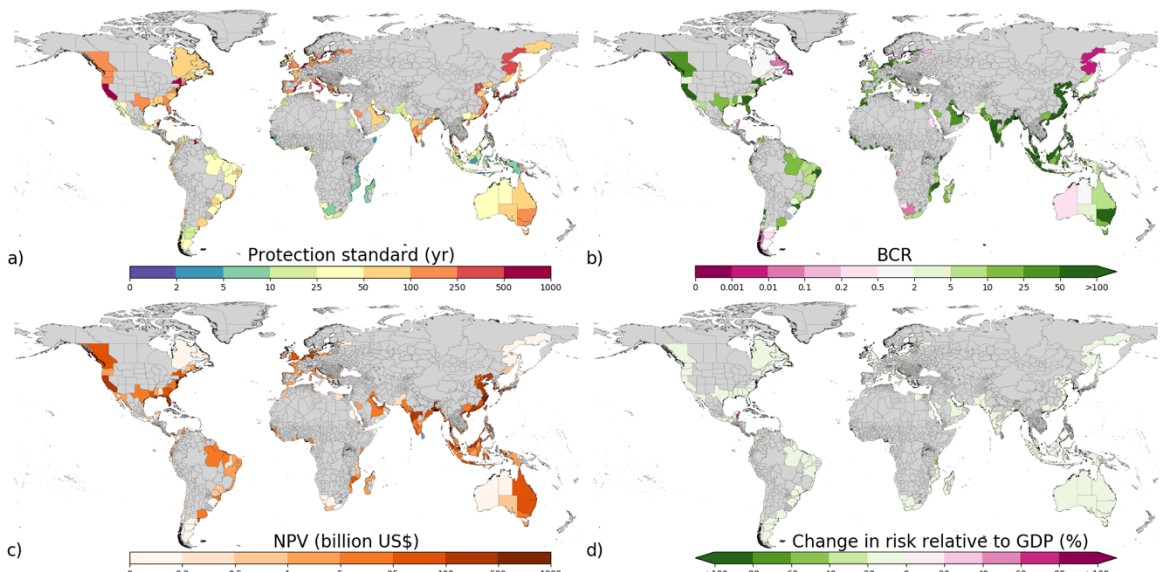

**Figure 7: 'Relative risk constant' adaptation objective results of (a) protection standards; (b) BCRs; (c) total NPV; and (d) change in risk relative to GDP for RCP4.5/SSP2.**

In the **'optimize'** adaptation objective (Figure 8), the highest optimal protection standards are generally found in East Asia, Southeast Asia, South Asia, and the Gulf coast of the USA. High protection standards are also found in parts of Europe and other parts of the USA, parts of western and eastern Africa, some parts of South America, and south-eastern Australia. The highest change in protection standards compared to current are found in South Asia and Southeast Asia. In most sub-national regions, the benefits exceed the costs when upgrading protection standards (89%). However, in some sub-national regions the BCR is less than 1 (indicated with hatched lines). The highest values of NPV (Figure 8c) are found in parts of South and Southeast Asia, North America, and northwest Europe. While most sub-national regions show a positive return on investment, there is still an increase in relative risk in 32% of the sub-national regions assessed, under the 'optimize' adaptation objective. Regions where this is especially the case include: Europe, North America, South America, Japan and Australia, as shown in Figure 8d. Many sub-national regions with decreases in relative risk can be found in South Asia, Southeast Asia, parts of the Gulf coast of the USA, New South Wales in Australia, several sub-national regions in Africa, and some parts of South America, among others. Generally, in these regions, protection standards and/or absolute dike heights increase the most.


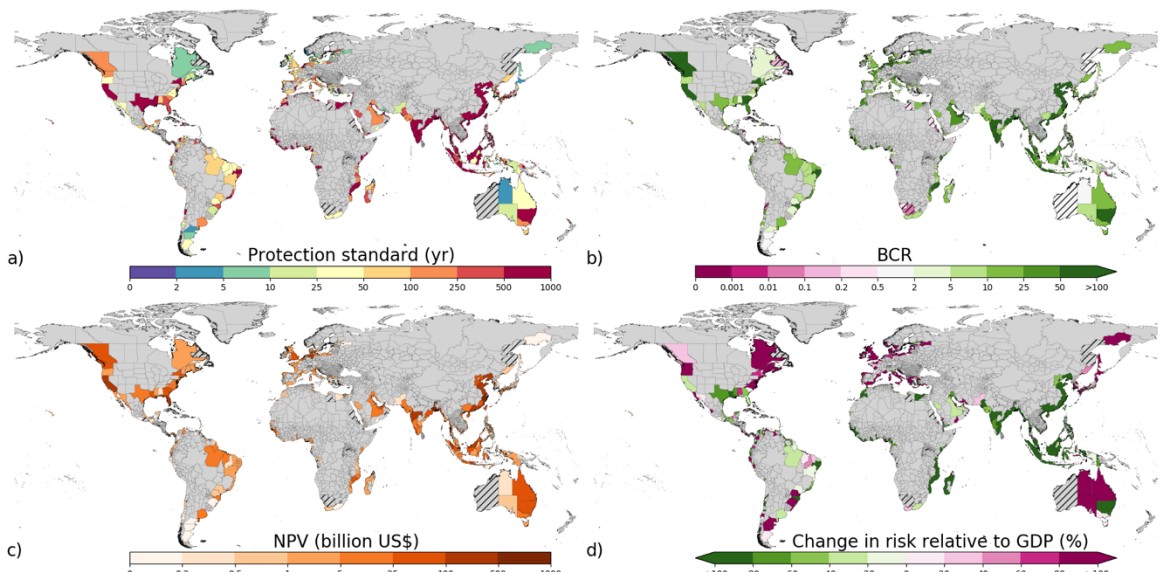

**Figure 8: 'Optimize' adaptation objective results of (a) optimal protection standards; (b) BCRs; (c) total NPV; and (d) change in risk relative to GDP for RCP4.5/SSP2. Regions where no optimal protection standards are found are indicated with hatched lines.**

### 3.4 Attribution of costs to different drivers of risk

In Figure 9, we show the percentage of the total costs of the 'optimize' adaptation objective (Figure 9a) that can be attributed to each of the following risk drivers: climate change (in this case sea-level rise) (Figure 9b); optimizing current protection standards (Figure 9c); socioeconomic change (Figure 9d); and subsidence (Figure 9e). The results are shown for the RCP4.5/SSP2 scenario and only for sub-national regions that have a BCR higher than 1 in the 'optimize' adaptation objective. The total costs exceed US$1 billion for 4% of the sub-national regions assessed and exceed US$1 million for 87%. For most parts of the globe, climate change (in this case sea level rise) contributes the most to the costs of adaptation, exceeding 50% of the total costs in 98% of the sub-national regions (Figure 9a), and exceeding 90% of the total costs in 58% of the sub-national regions. However, the other drivers can also play an important role, but are dwarfed in absolute terms by the costs related to sea-level rise. For example, in South Asia Southeast Asia, and East Africa optimizing to current conditions and socioeconomic change are important drivers and, in some cases, the most important driver. There are some other regional exceptions where climate change is not the most dominant driver of adaptation costs. Moreover, locally land subsidence due to groundwater extraction can cause huge flood problems and bring large costs in some areas (Dixon et al., 2006; Yin et al., 2013), but are not seen when aggregated to the sub-national regions of this study. However, there are a few regions where subsidence is a more dominant driver (i.e. parts of India, China, Japan, and Taiwan). The same patterns are found in the attribution results for the RCP8.5/SSP5 scenario, which can be found in the Supplementary Figure S6.



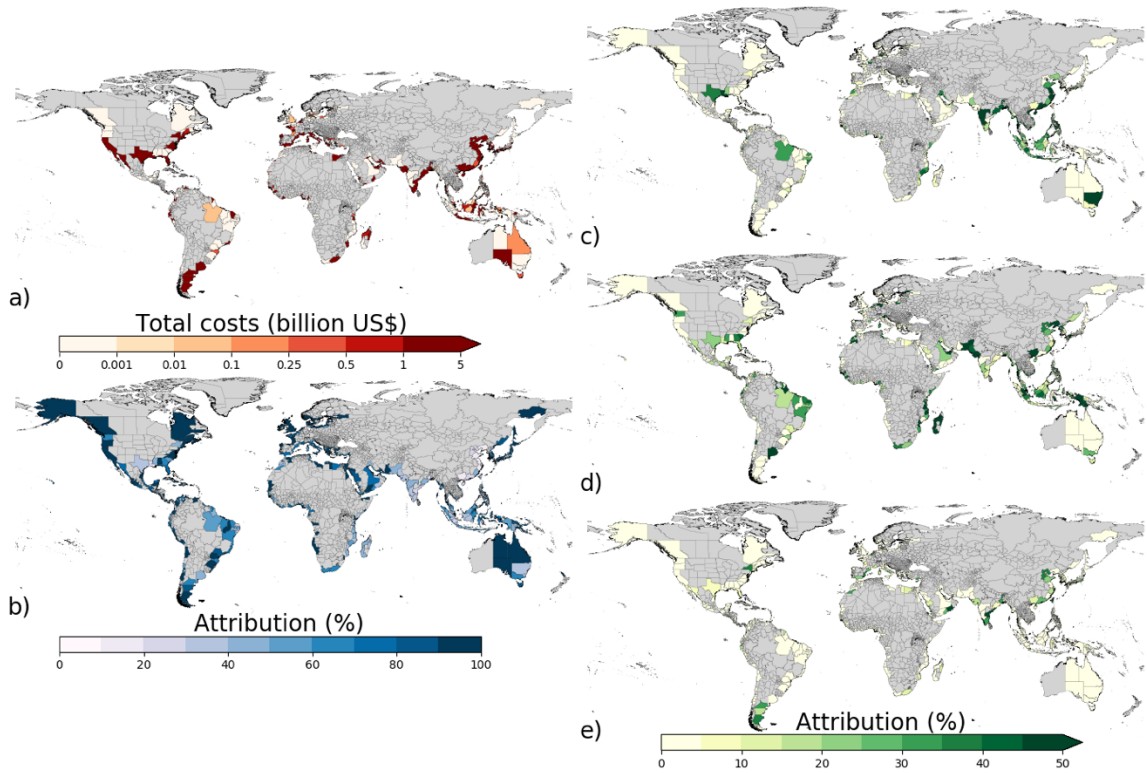

**Figure 9: Attribution of costs overview for RCP4.5/SSP2 with (a) total costs; (b) attribution of sea-level rise ($ATR_{SLR}$); (c) attribution of current optimizing ($ATR_{CUR}$); (d) attribution of socioeconomic change ($ATR_{SEC}$); and (e) subsidence ($ATR_{SUB}$). Note that the attribution of SLR is on a different scale.**

Figure 10 shows the attribution of the costs for the same scenario and adaptation objective, aggregated to the World Bank regions. In all the regions (except South Asia), sea-level rise is the most dominant driver, accounting for between 27% (South Asia) and 79% (Europe & Central Asia) of the costs of adaptation. The costs of increasing dike height to achieve optimal protection under current conditions are highest in the Global South. This is especially the case for the East Pacific & Asia and South Asia regions, with values of 22% and 38% respectively. The relative contribution of socioeconomic change is largest in East Asia & Pacific, South Asia and Sub-Saharan Africa, with values of 20%, 26% and 27% respectively. Of all drivers, subsidence is the least dominant with values up to 9% (East Asia & Pacific) and 10% (Middle East & North Africa). Supplementary Figure S7 shows the attribution aggregated to the World Bank regions for RCP8.5/SSP5.

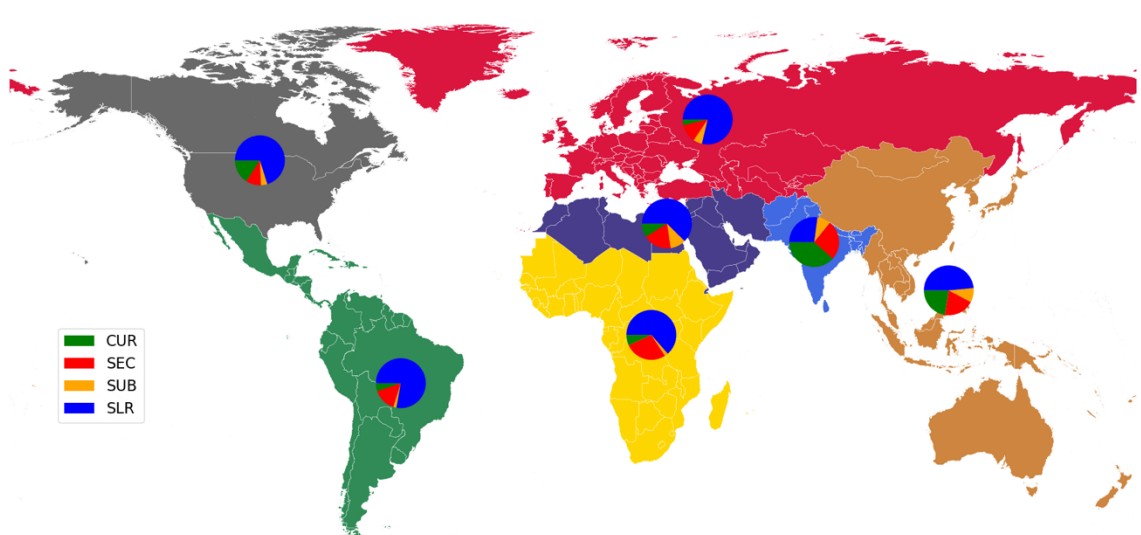

**400** **Figure 10: Attribution of costs of adaptation for World Bank regions under the 'optimize' adaptation objective and RCP4.5/SSP2 for optimizing to current conditions (CUR), socio-economic change (SEC) subsidence (SUB), and sea-level rise (SLR).**

### 3.5 Sensitivity analysis

In this section, we show the sensitivity of the results to the use of different: SSPs, sea-level rise projections, discount rates, and operation and maintenance (O&M) costs. In Table 2, we show results (of BCR) standardised to a baseline scenario with the

**405** following assumptions: RCP4.5, SSP2 (middle of the road), discount rate of 5%, and O&M of 1%. We employed a one-at-a-time sensitivity analysis, so for each row in the table only one parameter has changed, and the values shown are standardised by calculating the relative change. All associated BCRs for the standardised values shown in Table 2 are still higher than 1. Globally, BCRs range between 45 and 119 for the different model runs (73 for the reference). At the global scale the BCRs are most sensitive to the use of the different SSPs and discount rates. They cause the largest changes in BCR, with standardised

**410** values of 0.44 and 2.17 found in South Asia and Sub-Saharan Africa. Differences in SLR input affect the BCR by a factor of up to 0.38. Europe & Central Asia and North America are the least sensitive to the changes in input parameters. The O&M costs show BCRs that are more in line with the reference model run, higher or lower values up to 0.18.

**Table 2: Sensitivity analysis of model runs with different input parameters. BCRs are standardised to the model run with RCP4.5/SSP2, discount rate of 5%, and O&M costs of 1%.**

|  | East Asia & Pacific | Europe & Central Asia | Latin America & Caribbean | Middle East & North Africa | North America | South Asia | Sub-Saharan Africa | Global |
|---|---|---|---|---|---|---|---|---|
| **Reference BCR** | 90 | 99 | 13 | 77 | 29 | 199 | 35 | 73 |
| *Sensitivity to SSP projection* | | | | | | | | |





| | | | | | | | | |
|---|---|---|---|---|---|---|---|---|
| **SSP1** | 1.35 | 1.02 | 1.21 | 1.06 | 0.97 | 1.60 | 1.66 | 1.33 |
| **SSP3** | 0.66 | 0.88 | 0.73 | 0.75 | 0.76 | 0.45 | 0.45 | 0.65 |
| **SSP4** | 1.02 | 0.98 | 0.94 | 0.93 | 1.01 | 0.84 | 0.47 | 0.95 |
| **SSP5** | 1.70 | 1.11 | 1.52 | 1.26 | 1.19 | 2.15 | 2.20 | 1.64 |
| *Sensitivity to SLR projection* | | | | | | | | |
| **SLR low** | 1.07 | 1.38 | 1.06 | 1.04 | 1.11 | 1.00 | 1.13 | 1.13 |
| **SLR high** | 0.93 | 0.74 | 0.92 | 0.85 | 0.89 | 0.97 | 0.92 | 0.86 |
| *Sensitivity to discount rate* | | | | | | | | |
| **$r$ 3%** | 1.55 | 1.13 | 1.61 | 1.45 | 1.38 | 1.79 | 1.76 | 1.50 |
| **$r$ 8%** | 0.62 | 0.82 | 0.57 | 0.62 | 0.70 | 0.49 | 0.50 | 0.62 |
| *Sensitivity to O&M rate* | | | | | | | | |
| **O&M 0.1%** | 1.14 | 1.12 | 1.16 | 1.10 | 1.15 | 1.18 | 1.16 | 1.14 |
| **O&M 2%** | 0.88 | 0.88 | 0.89 | 0.86 | 0.86 | 0.86 | 0.86 | 0.88 |

## 3.6 Comparison to previous studies

Hallegatte et al. (2013) performed a study on future flood risk for 136 major coastal cities. They estimated an EAD of US$6 billion for current conditions, while in our study we find an EAD of US$19.6 billion. Our estimates of EAD is higher, which is to be expected given that we estimate risk for all coastlines as opposed to 136 major coastal cities. Hallegatte et al. (2013) projected future risk increasing up to US$60-63 billion if protection standards are kept constant by 2050. In our study we find an EAD of US$84 billion by 2050 when keeping protection standards constant (RCP4.5/SSP2 scenario). If no adaptation is implemented in 2050, Hallegatte et al. (2013) estimate EAD over US$1 trillion, where we find US$1.1 trillion.

Hinkel et al. (2010) attributed adaptation costs to sea-level rise using dikes for the European Union. They estimated this to be between US$2.6-3.5 billion. In our results we find values between US$12.9 billion and US$22.7 billion for the European Union for the scenarios RCP4.5 and RCP8.5 respectively. In a follow-up study, Hinkel et al. (2014) estimate global costs of protecting the coast with dikes. They estimate a range of US$12-71 billion, while our study estimates the global costs of adaptation for the 'optimize' adaptation objective between US$152 billion and US$208 billion for the RCP4.5/SSP2 and RCP8.5/SSP5 respectively. It should be noted that Hinkel et al. (2010) and Hinkel et al. (2014) use a demand function for safety where dikes are raised following relative sea-level rise and socioeconomic development, while we optimise protection standards by maximizing NPV. Moreover, the scenarios used in the studies are different.

A recent study by Lincke & Hinkel (2018) found that it is economically feasible to invest in protection for 13% of the coast globally. Using their method they found a lower share of protected coastline compared to previous studies (Nicholls et al., 2008; Tol, 2002). In our study, we found that for the 'optimize' adaptation objective 698 out of the 784 sub-national regions have a BCR higher than 1, indicating that it is economically feasible to implement adaptation in many regions through raising dikes. In our study, the benefit-cost analysis is carried out at the sub-national scale, whereby dikes are only raised on coastal reaches where our transects show there to be potential hazard (inundation) and urban exposure. If we calculate the percentage





of the entire global coastline for which this leads to dike heightening in our model with a BCR higher than 1, it amounts to 3.4% of the global coastline. This is lower than the value in Lincke & Hinkel (2018), but this can be explained by the fact that the distance between our transects is 1 kilometre horizontal resolution at the equator, whilst Lincke & Hinkel (2018) raise dikes along the coast of entire coastal segments, which have lengths ranging from 0.009 to 5213 kilometre, with a mean of 85 kilometre.

### 3.7 Limitations and future research

While our model scheme does not include dynamic inundation modelling, it does include resistance factors similar to those used by Vafeidis et al. (2019), in order to account for water-level attenuation. It therefore represents an advance on previous studies that have used planar inundation modelling methods (i.e. bathtub models). An improvement could be made by using a dynamic inundation modelling scheme (Vousdoukas et al., 2016), but at the cost of increased computing time. Another improvement can be made by including waves in our inundation modelling, which is found to be an important component in inundation modelling (Vousdoukas et al., 2017). The inundation modelling scheme can be further improved by increasing the resolution from 30" to a higher resolution in order to better understand local scale signals and patterns, since the scale of assessment and resolution of input data has a significant implication on flood risk model results (Wolff et al., 2016). However, we stress that this study aims to understand global flood risk and general patterns on the sub-national scale, and this study can be used as a first proxy indicating feasibility of adaptation through structural measures, such as dikes.

For this study, results are shown for the scenario RCP4.5/SSP2 and RCP8.5/SSP5 in supplementary information. The range of sea-level rise input values (between the 5th percentile of RCP4.5 and the 95th percentile of RCP8.5) cover a wide range of sea-level rise uncertainty (approximately 0.3 – 0.7m at the equator in the Atlantic Ocean). While in reality the effects of climate change will continue to rise beyond 2100 even if Paris Agreement is met (Clark et al., 2016), our study examines adaptation objectives until 2100. Results for all combinations of these two RCPs together with all five SSPs can be found in the supplementary data.

Several uncertainties exist on the cost calculation side. The first is the monetary value we assumed for the costs of dike heightening. Although we account for differences in costs between countries by using different construction factors and market exchange rates, in reality the costs might differ between regions and may be higher due to local conditions (both physical and socioeconomic). We also use a linear cost function for dike heightening. Using this linear cost function for large scale studies has been found a reasonable assumption according to (Lenk et al., 2017).

Another important uncertainty in this study is the current protection standards estimated with the FLOPROS modelling approach, as data on flood protection along the global coastlines are not available. These only provide a first order estimate of current protection standards per sub-national region. In the Supplementary Figure S8, a validation of the coastal protection standards estimated with the FLOPROS modelling approach is provided. Values are shown for several locations for which reliable reported estimates of protection standards are available. These reported values are either shown as a range (minimum and maximum reported values) or a single value. Overall, the model performs well. The only location for which the reported values provide a range, and the FLOPROS model lies outside this range, is Durban. However, note that reported values are for



the city of Durban, whilst the FLOPROS model value is for the state in which it is located. An improvement to this study could be made by, for instance, mapping flood protections globally by using Earth Observation-based methods.

In this study, several uncertainties exist with assumptions on expected damages per occupancy type. First, we assumed the percentage of occupancy type per grid cell to be the same for all location, whilst in reality it is spatially heterogeneous, and secondly, we assumed the building density per occupancy type. An improvement could be made by using Machine Learning

to improve accuracy of urban land cover and building types (Hecht et al., 2015; Huang et al., 2018). We also used depth-damage curves per occupancy type, but in reality, these curves also differ between buildings in these occupancy types. To further improve the exposure data of our framework, the Global Human Settlement layer can be used for high-resolution population mapping.

The sub-national regions where no adaptation objective shows a positive BCR, should not be interpreted that no adaptation to

coastal flood risk should take place. In fact, other adaptation measures (or a combination of multiple measures) besides raising dikes might be more economically feasible in any regions studied, including those with BCRs higher than 1. In this study we only assumed grey infrastructure as adaptation measures, but there are also other measures to reduce flood risk. For instance, the vulnerability can be improved by wet or dry proofing buildings (Aerts et al., 2014) or people and assets can be moved to less flood prone areas in order to reduce the exposure to floods (McLeman and Smit, 2006). Lastly, several local studies show

the benefits of nature-based or hybrid adaptation measures (Cheong et al., 2013; Jongman, 2018; Temmerman et al., 2013). Vegetation on the foreshore has a significant role in the breaking of waves (Shepard et al., 2011) and attenuates storm water levels (Zhang et al., 2012). An improvement could be made by including other adaptation measures besides grey infrastructure as adaptation measures.

## 4 Conclusion

In this study, four adaptation objectives for reducing (future) coastal flood risk through structural measures have been explored and a benefit-cost analysis has been performed on the sub-national scale for the entire globe. Furthermore, the costs of adaptation have been attributed to different drivers of flood risk: sea-level rise, socio-economic change, subsidence, and optimizing to current conditions. Globally, we find that EAD increases by a factor of 150 between 2010 and 2080, if we assume that no adaptation takes place, and find that 15 countries account for approximately 90% of this increase.

We find that all four adaptation objectives show high potential to reduce (future) coastal flood risk at the global scale in a cost-effective manner. The 'optimize' adaptation objective shows the highest NPV (more than US$11 trillion) with a BCR of 76, while the 'protection constant' adaptation objective shows the lowest NPV (US$9.5 trillion) with a BCR of 67 for the RCP4.5/SSP2 scenario.

At the regional scale, we show that the adaptation objectives can be achieved with a BCR more than 1 for most of the sub-

national regions. This ranges from 89% for the 'optimize' adaptation objective to 79% for the 'absolute risk constant' adaptation objective. However, we also show that under the 'optimize' adaptation objective, relative risk would still increase compared to current values in 32% of the sub-national regions assessed.



We assess the sensitivity of the results by performing a one-at-a-time sensitivity analysis to various assumptions and find that, given the uncertainties, implementing structural adaptation measures is a feasible solution to reduce (future) coastal flood risk.

Although differences in BCR exist, we show that changes in parameters still result in positive BCRs (between 45 and 120 globally) for the 'optimize' adaptation objective.

Attributing the total costs for the 'optimize' adaptation objective, we find that sea-level rise contributes the most and exceeds 50% of the total costs in 98% of the sub-national regions assessed, and exceeds 90% of the total costs in 58% of the sub-national regions. However, the other drivers also play an important role, but are dwarfed in absolute terms by the total costs

related to the attribution.

The results of this study can be used to highlight potential savings through adaptation at the sub-national scale. Clearly, local scale models and assessments should be used for the design and implementation of individual adaptation measures, but our results can be used as a first proxy indicating regions where adaptation through structural measures may be economically feasible. To increase the accessibility of the results to the risk community, results of this study will be integrated into the

Aqueduct Global Flood Analyzer webtool ([www.wri.org/floods](www.wri.org/floods)).

*Supplementary data availability.* The results of this study for all RCP and SSP combinations for protection standards, change in risk relative to GDP, B:C ratio and NPV for all four adaptation objectives are available at: [https://doi.org/10.5281/zenodo.3475120](https://doi.org/10.5281/zenodo.3475120). Figures of the results of RCP8.5/SSP5 combination are available at the

supplementary material.

*Author contribution.* TT, PJW and HdM conceived the study. All co-authors contributed to the development and design of the methodology. TT analysed the data and prepared the manuscript with contributions from all co-authors.

*Competing interest.* The authors declare no conflict of interest.

*Acknowledgement.* The research leading to these results received funding from: The Netherlands Organisation for Scientific Research (NWO) in the form of a VIDI grant (grant no. 016.161.324); the Aqueduct Global Flood Analyzer project, via subsidy 5000002722 from the Netherlands Ministry of Infrastructure and Water Management - the latter project is convened by the

World Resources Institute; and the Future Water Challenges 2 project, funded by the Netherlands Ministry of infrastructure and Water Management.

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
