# Peer review of "Global scale benefit-cost analysis of coastal flood adaptation to different flood risk drivers using structural measures"

_Natural Hazards and Earth System Sciences, 2019_

## Referee Comment (RC1) · Anonymous Referee #1 · 6 Jan 2020

**Comments**

General comments

The manuscript "Global scale benefit-cost analysis of coastal flood adaptation to different flood risk drivers" assesses the benefits and costs of four structural adaptation objectives at global scale until 2080. It further attributes the contribution of different flood risk drivers to the total adaptation costs under the 'Optimize' adaptation objective. For this analysis, the authors first assess coastal flood risk expressed in Expected Annual Damages (EAD), followed by the estimation of adaptation costs, before conducting a benefit-cost analysis (BCA) for each adaptation objective. The study finds that all adaptation objectives have a high potential for reducing flood risk in a cost-effective manner; further, the contribution of sea-level rise (SLR) dominates adaptation costs in most regions.

The study provides first estimates of the benefits of different structural adaptation objectives, taking into account a range of SLR and socioeconomic scenarios. It uses well-established methods and data and extends these for the purpose of this study, therefore providing new insights into the cost-effectiveness of adaptation strategies at the global scale. However, the manuscript in its current form has a number of limitations and I therefore propose to reconsider the manuscript for publication upon revision of the following issues:

Specific comments

1. As the study accounts for structural adaptation measures only, I would suggest adding this piece of information to the title of the manuscript.
2. While the introduction section cites the relevant background literature regarding coastal flood risk assessments, the current research gap is not pointed out clearly (l. 49-55). Consequently, the innovative aspects of this study do not become entirely clear. Similarly, previous work that has accounted for subsidence in assessing coastal food risk has not been cited (e.g. Hinkel et al. 2014, Nicholls et al. 2008, Hallegatte et al. 2013). Therefore, I suggest adding more detail to the respective sections.
3. L. 95: Please elaborate where the enriched GTSR data were acquired and how they were extended.
4. L. 116-119: How have the SLR projections been regionalized? Please provide more information.
5. To assess current exposure, you refer to the methodology of Huizinga et al. 2017. It remains unclear how exactly damages have been assessed without consulting the study of Huizinga et al. 2017. Please provide sufficient detail.
6. It is not clear to me why the HYDE database was used to assess current exposure as it has a coarse resolution and is rather outdated. In the discussion section (l. 477), the Global Human Settlement Layer (GHSL) is mentioned, which provides built-up land data of 2015 at resolutions of 30m, 250m, and 1km. Further, the GHSL data provide spatial population distributions at resolutions of 250m, 1km, 9 arcsecs, 30 arcsecs (https://ghsl.jrc.ec.europa.eu/download.php?ds=pop). Both GHSL datasets could be used in combination for assessing current exposure, which would increase consistency of the results while avoiding the use of correction factors if base year data do not align (l. 148-149).
7. The SSPs are introduced rather abruptly in l. 147, but further details are missing. Please provide a brief description of the SSPs along with the relevant literature (e.g. O'Neill et al., 2014; O'Neill et

al., 2017; van Vuuren et al., 2014). These pieces of information are also important to contextualize the results of the study (see also comment 18).

8. Some data for assessing exposure were downloaded from the SSP database, while others were not (e.g. GDP values). As the SSPs are the current state-of-the-art socioeconomic scenarios, I suggest using the national-level population projections as well as the GDP projections from the SSP database for the entire study period. Furthermore, spatial population projections based on the SSPs are available from Jones and O'Neill, 2016 at a resolution of 1/8 degree, downscaled to 30 arcsecs by Gao, 2017, and from Merkens et al., 2016, also at a resolution of 30 arcsecs. These may serve as a suitable basis for producing future simulations of built-up land, using the methodology of Winsemius et al 2016 (l. 151-153).

9. Section 2.1.5 provides a description of the results of FLOPROS rather than how the modeling approach was applied. I suggest stating the use of the FLOPROS data, and moving further explanation to the SI.

10. The scenario combinations (RCPs-SSPs) used for the analysis are briefly described in the results section (l. 292-296). I suggest moving the reasoning for using these scenario combinations to the methods section, along with additional background information.

11. Figure 3: It would be helpful if the results were contextualized in the text with regard to the respective drivers contributing to coastal flood risk under current and future conditions. Please also provide the country names for each ISO code.

12. In Figure 2 and Figures 5-8, a legend of the regions in gray color (i.e. no data?) is missing. Further, the scalebar of the BCR plot (panel b) does not allow for differentiating between BCRs > 1 and < 1. Additionally, the scalebar of the NPV plot (panel c) does not provide a signature for NPV = 0. The same holds true for panels b-e in Figure 9. I suggest adjusting the figures accordingly in order to increase the information conveyed by the figures. Furthermore, the administrative units in South Africa and Namibia (all panels) seem odd as they include areas of Botswana, which is a landlocked country. Please also revise the administrative unit data.

13. Figure 8: It would be interesting if the change in risk (panel d) was contextualized in more detail, providing explanations of increases and decreases in flood risk in the text (see also comment 18).

14. Figure 10: Some of the colors used for the World Bank regions are misleading as they align with those used for the flood risk drivers. Please revise the colors used.

15. Table 2: You mention in l. 120 that the 5$^{th}$ and 95$^{th}$ percentiles of the SLR projections are used for the sensitivitiy analysis. Do SLR low and SLR high refer to these percentiles?

16. Section 3.6 provides useful insights into the results of other studies, but lacks detailed explanation of the reasons for differences between this study and previous work. The results of this study are considerably higher than those of previous work despite the more refined inundation modeling approach used. I would suggest extending this section accordingly, by providing more context.

17. L. 477 please provide a reference for the GHSL data.

18. Contextualization of the results is largely missing in the discussion section (see also comments 7 and 16). It would be helpful for the reader if the different adapation objectives were discussed in more detail, addressing questions such as: What do different adaptation objectives mean/entail? Which would be more desirable based on the BCRs? Why does flood risk increase in certain regions under certain objectives (see also comment 13)? I suggest adding a section that elaborates these aspects to the discussion. Connected to this point, it would also be insightful if the benefits of the study were elaborated in more detail, for instance how other scholars and/or decision-makers could use the results.

Technical corrections

19. List of typos/mistakes found:
    - L. 26: 'compared to' stated twice
    - L. 27: remove '.' after Raftery et al. (2017)
    - L. 100: hydrologically
    - L. 129: 30" x 30"
    - L. 212: remove ',' after Jevrejeva et al. 2014
    - L. 380: add ',' after South Asia
    - L. 473: locations
20. The manuscript uses British English and American English interchangeably, one example being 'optimize', 'optimise', 'optimisation' etc in section 2.3.

**References** (in addition to those already cited in the manuscript)

Gao, J.: Downscaling Global Spatial Population Projections from 1/8-degree to 1-km Grid Cells, NCAR Technical Note, NCAR/TN-537+STR, 2017.

Jones, B. and O'Neill, B. C.: Spatially explicit global population scenarios consistent with the Shared Socioeconomic Pathways, Environ. Res. Lett., 11, 84003, doi:10.1088/1748-9326/11/8/084003, 2016.

Merkens, J.-L., Reimann, L., Hinkel, J., and Vafeidis, A. T.: Gridded population projections for the coastal zone under the Shared Socioeconomic Pathways, Global and Planetary Change, 145, 57–66, doi:10.1016/j.gloplacha.2016.08.009, 2016.

O'Neill, B. C., Kriegler, E., Ebi, K. L., Kemp-Benedict, E., Riahi, K., Rothman, D. S., van Ruijven, B. J., van Vuuren, D. P., Birkmann, J., Kok, K., Levy, M., and Solecki, W.: The roads ahead: Narratives for shared socioeconomic pathways describing world futures in the 21st century, Global Environmental Change, 42, 169–180, doi:10.1016/j.gloenvcha.2015.01.004, 2017.

O'Neill, B. C., Kriegler, E., Riahi, K., Ebi, K. L., Hallegatte, S., Carter, T. R., Mathur, R., and van Vuuren, D. P.: A new scenario framework for climate change research: The concept of shared socioeconomic pathways, Climatic Change, 122, 387–400, doi:10.1007/s10584-013-0905-2, 2014.

van Vuuren, D. P., Kriegler, E., O'Neill, B. C., Ebi, K. L., Riahi, K., Carter, T. R., Edmonds, J., Hallegatte, S., Kram, T., Mathur, R., and Winkler, H.: A new scenario framework for Climate Change Research: Scenario matrix architecture, Climatic Change, 122, 373–386, doi:10.1007/s10584-013-0906-1, 2014.

---

## Referee Comment (RC2) · Anonymous Referee #2 · 7 Jan 2020

Overall this is an interesting paper. The approach undertaken is robust and I commend the authors for their nice study. The work builds on several previous assessments, and presents an incremental step forward, rather than a step change. However, I think it has some novel elements and is certainly worthy of publication in NHESS and results will be of interest to many. I have listed 5 modest corrections that I would like to see addressed and several minor ones.

Modest corrections:

In lines 50-55, you discuss the previous studies, and then go onto say what the objectives of your paper are. I think you need to make it clearer how your paper is distinct from these previous assessments. At the moment this does not come across strongly enough.

[Figure]

Please provide, on lines 94 to 99, more details of how exactly you have included the tropical cyclones. Over what period was this done? How did you covert along track data into spatially varying wind and pressure fields?

Lines 124 to 133: I am not clear if these subsidence rates include glacial isostatic adjustment or not. Do they? Can you make this clear. I assume you are accounting for these effects. If not, then it significantly undervalues your results.

Lines 172 – 185: I found the description of the protection standards confusing. Please can you improve this section. Has this approach me validated, in regions for example, where the protect standards are known exactly. How does these compare to what Hallegate et al (2013) used in coastal cites? You cite the Netherlands as havimg a value of 1000. What are the units? Years? Please add these.

Why is your analysis based on 2080, and not 2100? TO me, it would seem much more sensible to go to 2100?

Minor corrections:

Line25 – I would maybe update to the special IPCC report in 2019, which is a bit more up to date.

Line 27 – there is an extra full stop after the Raftery reference.

Line 29 – you could add 'and change in in tides.

---

## Author Comment (AC1) · 14 Feb 2020

**Response to referee 1**

Manuscript for Natural Hazards and Earth System Sciences

**Title: Global scale benefit-cost analysis of coastal flood adaptation to different flood risk drivers using structural measures**

**Authors**

Timothy Tiggeloven, Hans de Moel, Hessel C. Winsemius, Dirk Eilander, Gilles Erkens, Eskedar Gebremedhin, Andres Diaz Loaiza, Samantha Kuzma, Tianyi Luo, Charles Iceland, Arno Bouwman, Jolien van Huijstee, Willem Ligtvoet, Philip J. Ward

**General response**
We would like to thank referee 1 for the time taken to critically review our manuscript. We are very pleased that the referee finds the manuscript to be providing new insights on the topic of adaptation strategies at the global scale. The referee raised a number of specific comments and technical corrections. One of the most important is with regards to contextualization of the results, which was largely missing in the manuscript. Based on this, we have included a section dedicated to the contextualization and included information in the text to strengthen this. The referee also comments on the need to more explicitly state the innovative aspects and distinction of this study from previous assessments. We have addressed this concern along with detailed comments about the approach and methodology of this assessment. We believe that these revisions to the manuscript, and those detailed below, have led to a significant improvement in our manuscript. In the following pages, we respond to the comments of the referee point by point. Our responses are shown in italics.

**Referee: 1**
General comments

The manuscript "Global scale benefit-cost analysis of coastal flood adaptation to different flood risk drivers" assesses the benefits and costs of four structural adaptation objectives at global scale until 2080. It further attributes the contribution of different flood risk drivers to the total adaptation costs under the 'Optimize' adaptation objective. For this analysis, the authors first assess coastal flood risk expressed in Expected Annual Damages (EAD), followed by the estimation of adaptation costs, before conducting a benefit-cost analysis (BCA) for each adaptation objective. The study finds that all adaptation objectives have a high potential for reducing flood risk in a cost-effective manner; further, the contribution of sea-level rise (SLR) dominates adaptation costs in most regions.

The study provides first estimates of the benefits of different structural adaptation objectives, taking into account a range of SLR and socioeconomic scenarios. It uses well-established methods and data and extends these for the purpose of this study, therefore providing new insights into the costeffectiveness of adaptation strategies at the global scale. However, the manuscript in its current form has a number of limitations and I therefore propose to reconsider the manuscript for publication upon revision of the following issues:

- *Many thanks for the encouraging words. We are very pleased that the referee finds the manuscript to provide new insights into the cost-effectiveness of adaptation objectives at the global scale.*

Specific comments

1. As the study accounts for structural adaptation measures only, I would suggest adding this piece of information to the title of the manuscript.

- *Thank you: we have added 'using structural measures' to the title of the manuscript.*

2. While the introduction section cites the relevant background literature regarding coastal flood risk assessments, the current research gap is not pointed out clearly (l. 49-55). Consequently, the innovative aspects of this study do not become entirely clear. Similarly, previous work that has accounted for subsidence in assessing coastal food risk has not been cited (e.g. Hinkel et al. 2014, Nicholls et al. 2008, Hallegatte et al. 2013). Therefore, I suggest adding more detail to the respective sections.

- *Thank you: we have addressed this in the revised manuscript by adding information about the innovative aspects of this study. We have also included the references with regards to subsidence in assessing coastal flood risk in previous assessments. Now this section reads: 'Recent studies have shown that adaptation measures hold a large potential for significantly reducing this future flood risk (Diaz, 2016; Hinkel et al., 2014; Lincke and Hinkel, 2018). However, the number of global scale studies in which the benefits and costs of disaster risk reduction and adaptation are explicitly and spatially accounted for remains limited. Existing studies have assessed the effect of climate change, subsidence and/or socioeconomic change (Hallegatte et al., 2013; Hinkel et al., 2014; Nicholls et al., 2008; Vousdoukas et al., 2016), but have not included adaptation objectives or attributed flood risk drivers to adaptation costs. Lincke & Hinkel (2018) assessed the cost-effectiveness of structural protection measures against sea-level rise and population growth using the DIVA model. They found that structural adaptation measures are for 13% of the global coastline feasible to invest in. However, they did not include subsidence and attribution of drivers in their modelling scheme.*
  *In this paper, we develop a model to evaluate the future benefits and costs of structural adaptation measures at the global scale. We use it to address the limitations of current studies addressed above, and thereby extend the current knowledge on the cost-effectiveness of structural adaptation measures in several ways. Firstly, we include subsidence due to groundwater extraction. Secondly, we assess the benefits and costs of several adaptation objectives. Thirdly, we attribute the costs of adaptation to different drivers (namely sea-level rise, subsidence and change in exposure).'*

3. L. 95: Please elaborate where the enriched GTSR data were acquired and how they were extended.

- *Thanks – we have included the following sentences to the manuscript in order to elaborate on the enrichment of GTSR with regards to tropical cyclone tracks: 'All tracks over the period 1979-2004 are used and converted into wind and pressure fields using the parametric Holland model (Delft3D-WES, 2019) in order to simulate aaalternative water levels using GTSM. These water levels are combined with the time series of GTSR by using the highest water level at each GTSM cell for each time step. Extreme values are estimated using a Gumbel extreme value distribution fit on the annual extremes.'*

4. L. 116-119: How have the SLR projections been regionalized? Please provide more information.

- *Thanks – We have added this piece of information in section 2.1.1 to provide information about the regionalization of the SLR projections:* 'We use gridded datasets of regional sea-level rise estimates developed by Jackson and Jevrejeva (2016). These data were derived by combining spatial patterns of individual sea-level rise contributions in a probabilistic.'

5. To assess current exposure, you refer to the methodology of Huizinga et al. 2017. It remains unclear how exactly damages have been assessed without consulting the study of Huizinga et al. 2017. Please provide sufficient detail.

- *We have included additional information in the manuscript, which clarifies the study without consulting it. It now reads:* 'Current maximum economic damages are estimated using the methodology of Huizinga et al. (2017). They used a root function to link GDP per capita to construction costs for each country. To convert construction costs to maximum damages, several adjustments are carried out using the suggested factors by Huizinga et al. (2017) for the different occupancy types. Such factors include depreciation and undamageable parts of buildings As a proxy for an approximation of percentage area per occupancy type, we set the urban grid cells of the layers from the HYDE database to 75% residential, 15% commercial and 10% industrial, based on a study by (BPIE, 2011) and a comparison of European cities' share of occupancy type of the CORINE Land Cover data (EEA, 2016). Following Huizinga et al. (2017), the density of buildings per occupancy types are set to 20% for residential and 30% for commercial/industrial.'

6. It is not clear to me why the HYDE database was used to assess current exposure as it has a coarse resolution and is rather outdated. In the discussion section (l. 477), the Global Human Settlement Layer (GHSL) is mentioned, which provides built-up land data of 2015 at resolutions of 30m, 250m, and 1km. Further, the GHSL data provide spatial population distributions at resolutions of 250m, 1km, 9 arcsecs, 30 arcsecs (https://ghsl.jrc.ec.europa.eu/download.php?ds=pop). Both GHSL datasets could be used in combination for assessing current exposure, which would increase consistency of the results while avoiding the use of correction factors if base year data do not align (l. 148-149).

- *Thank you for the suggestion. We use the HYDE database to keep consistency between current and future built-up area data, because it provides current and future landuse data . We believe that having a consistency in the dataset between future and current exposure data improves the robustness of the produced results. It is true that the HYDE database is a coarse resolution, but to our knowledge no such dataset yet exists given the mentioned requirements.*

7. The SSPs are introduced rather abruptly in l. 147, but further details are missing. Please provide a brief description of the SSPs along with the relevant literature (e.g. O'Neill et al., 2014; O'Neill et al., 2017; van Vuuren et al., 2014). These pieces of information are also important to contextualize the results of the study (see also comment 18).

- *Thank you: we have included the following information in the future exposure section of 2.1.2 in the revised manuscript:* 'These simulations include five narrative descriptions of future societal development associated with SSP1-5 (O'Neill et al., 2014). Such descriptions include

sustainability associated with low challenges (SSP1), middle of the road associated with intermediate challenges (SSP2), regional rivalry associated with high challenges (SSP3), inequality associated with dominance of adaptation challenges (SSP4) and Fossil-fueled development where the mitigation challenges are dominating (SSP5) (O'Neill et al., 2017).'

8. Some data for assessing exposure were downloaded from the SSP database, while others were not (e.g. GDP values). As the SSPs are the current state-of-the-art socioeconomic scenarios, I suggest using the national-level population projections as well as the GDP projections from the SSP database for the entire study period. Furthermore, spatial population projections based on the SSPs are available from Jones and O'Neill, 2016 at a resolution of 1/8 degree, downscaled to 30 arcsecs by Gao, 2017, and from Merkens et al., 2016, also at a resolution of 30 arcsecs. These may serve as a suitable basis for producing future simulations of built-up land, using the methodology of Winsemius et al 2016 (l. 151-153).

- *Thanks. We use gridded GDP layers from Van Huijstee et al. (2018) that uses GDP estimates from the SSP database as input to create those layers. This means that we have a consistency between the SSP data we use throughout our methodology. We see that we have not made this clear in the manuscript, so we adjusted the future exposure section in 2.1.2, which now reads:* 'In order to calculate future risk relative to GDP, future gridded GDP values are taken from Van Huijstee et al. (2018), which uses the national GDP per capita from the SSP database as input'.

9. Section 2.1.5 provides a description of the results of FLOPROS rather than how the modeling approach was applied. I suggest stating the use of the FLOPROS data, and moving further explanation to the SI.

- *Thank you for the suggestion. We have amended this information to the SI. In the main manuscript it now reads:* 'In order to assess the benefits and costs of adaptation objectives, information on current protection standards is needed. We use the FLOPROS modelling approach (Scussolini et al., 2016) to estimate these protection standards using current exposure data and EAD data from the GLOFRIS model as input. Figure 2d shows the estimated FLOPROS flood protection standards for each coastal sub-national unit. Further information about the FLOPOS estimates together with a validation of the results can be found in the Supplementary Information.'
- *And in the supplementary information it now reads:* 'This section contains a brief description of the coastal protection standards estimated with the FLOPROS modelling approach. Higher protection standards can be found at regions with high economic activity and high asset exposure. Regions with low risk have lower estimated protection standards. Regions without modelled risk in the GLOFRIS model are excluded. This occurs in regions where we have no data on exposure or no coastal inundation is simulated. These protection standards are used in our paper as the current protection, on top of which the future costs of dike heightening are calculated. The protection standards for The Netherlands are manually set to 1000-year return period. This is because, for whole of The Netherlands protection standards are known to be higher than 1000-year return period.'

10. The scenario combinations (RCPs-SSPs) used for the analysis are briefly described in the results section (l. 292-296). I suggest moving the reasoning for using these scenario combinations to the methods section, along with additional background information.

- *Amended. We have moved the information together with some additional information to the methods section, specifically section 2.3. It now reads: 'The benefit-cost analysis is carried out for two different sea-level rise scenarios (RCPs) and five different socioeconomic scenarios (SSPs). All the results are shown for two scenario combinations (van Vuuren et al., 2014), namely RCP4.5/SSP2 and RCP8.5/SSP5. The former is used for a 'middle of the road' scenario with medium challenges and adaptation (Riahi et al., 2017) that can broadly be aligned with the Paris agreement targets (Tribett et al., 2017), while the latter is used as a 'fossil-fuel development' world (Kriegler et al., 2017). Results of the other combinations can be found in the supplementary data.'*

11. Figure 3: It would be helpful if the results were contextualized in the text with regard to the respective drivers contributing to coastal flood risk under current and future conditions. Please also provide the country names for each ISO code.

- *Thanks – we have included the following information in section 3.4 in order to contextualize the drivers contributing to coastal flood risk: 'The results show that climate change is not the most dominant driver in 4 of the 5 countries that have the highest share of future EAD if no adaptation takes place (e.g. China, Bangladesh, India, and Indonesia)'. Also, we have included information about the ISO codes we used in the caption of the corresponding figures (ISO 3166-1 alpha-3 codes).*

12. In Figure 2 and Figures 5-8, a legend of the regions in gray color (i.e. no data?) is missing. Further, the scalebar of the BCR plot (panel b) does not allow for differentiating between BCRs > 1 and < 1. Additionally, the scalebar of the NPV plot (panel c) does not provide a signature for NPV = 0. The same holds true for panels b-e in Figure 9. I suggest adjusting the figures accordingly in order to increase the information conveyed by the figures. Furthermore, the administrative units in South Africa and Namibia (all panels) seem odd as they include areas of Botswana, which is a landlocked country. Please also revise the administrative unit data.

- *Thank you for the suggestions. We have adjusted the scalebar so that NPV below 0 is indicated as a separate group. Also, we have included information in the captions of the figures with regards to the grey colour and no data. We have not adjusted the differentiation between BCR 0.5 and 2, because in practice decision-makers will not implement a large scale project when the BCR is just over (or under) 1. Therefore, we have decided that because of this and the uncertainty of future projections, we keep the original BCR categories. However, all data for all units can be found in the data files in the Supplementary Data. With regards to the administrative units, we have double-checked the units and find that they are correct and the borders between South Africa, Namibia and Botswana are preserved in our figures.*

13. Figure 8: It would be interesting if the change in risk (panel d) was contextualized in more detail, providing explanations of increases and decreases in flood risk in the text (see also comment 18).

- *Thanks – we have included the following sentences in section 3.3 focussing on the 'optimize' adaptation objective: 'While most sub-national regions show a positive return on investment, there is still an increase in relative risk in 32% of the sub-national regions assessed, under the 'optimize' adaptation objective. In these cases, it is economically efficient to implement protection measures up to a certain level, yet the economic costs of keeping EAD as a*

percentage of GDP constant would exceed the avoided damages. Regions where this is especially the case include: Europe, North America, South America, Japan and Australia, as shown in Figure 8d. Many sub-national regions with decreases in relative risk can be found in South Asia, Southeast Asia, parts of the Gulf coast of the USA, New South Wales in Australia, several sub-national regions in Africa, and some parts of South America, among others. In these regions, the increase in risk is generally very high, which means that the costs of investment in protection are lower than the avoided damages relative to GDP. Generally, in these regions, protection standards and/or absolute dike heights increase the most.'

14. Figure 10: Some of the colors used for the World Bank regions are misleading as they align with those used for the flood risk drivers. Please revise the colors used.

- *Amended – we revised the colours used.*

15. Table 2: You mention in l. 120 that the 5th and 95th percentiles of the SLR projections are used for the sensitivitiy analysis. Do SLR low and SLR high refer to these percentiles?

- *Indeed. We have clarified this in the caption of Table 2.*

16. Section 3.6 provides useful insights into the results of other studies, but lacks detailed explanation of the reasons for differences between this study and previous work. The results of this study are considerably higher than those of previous work despite the more refined inundation modeling approach used. I would suggest extending this section accordingly, by providing more context.

- *Thank you – in this section we compare our results to three different studies that use similar approaches as our study. Firstly, we find that we simulate higher values for current global EAD than Hallegatte et al. (2013) and reason that this can be attributed to the extent of their study. They use 136 major coastal cities, whilst we use all global coastlines. For future simulations of 2050 our values are in the same range of Hallegatte et al. (2013). Then, we compare our findings of adaptation costs to the findings of Hinkel et al. (2010) and Hinkel et al. (2014), and find that our results are higher than their findings. We reason that it should be noted that they use a demand-function for adaptation, while we do not use that function, but rather maximize NPV. This adaptation objective allows dynamic optimization per sub-national region and can result in higher adaptation costs as long as the net benefits increase. Additionally, we use different scenarios than those used in Hinkel et al. (2010) and Hinkel et al. (2014). Lastly, we compare our results of economic feasibility for sub-national regions and coastlines to the findings of Lincke & Hinkel (2018). Although in 89% of sub-national regions assessed in this study it is economically feasible to adapt, we find that the total coastline that is protected in these sub-national regions amount up to 3.4% of global coastline, compared to 13% of the coast globally that Lincke & Hinkel (2018) find. We reason that this difference is a result of the difference in spatial aggregation. They optimize transects ranging from 0.009 to 5213 kilometre with a mean of 85 kilometre, while we use 1 kilometre horizontal resolution at the equator. This can explain why we have lower percentage of coast that is feasible to protect than Lincke & Hinkel (2018).*

17. L. 477 please provide a reference for the GHSL data.

- *Amended.*

18. Contextualization of the results is largely missing in the discussion section (see also comments 7 and 16). It would be helpful for the reader if the different adapation objectives were discussed in more detail, addressing questions such as: What do different adaptation objectives mean/entail? Which would be more desirable based on the BCRs? Why does flood risk increase in certain regions under certain objectives (see also comment 13)? I suggest adding a section that elaborates these aspects to the discussion. Connected to this point, it would also be insightful if the benefits of the study were elaborated in more detail, for instance how other scholars and/or decision-makers could use the results.

- *Thank you. We agree that the contextualization can be improved. Therefore we added a section dedicated to the contextualization along with additional information in section 3.3: 'In the middle of the road scenario of RCP4.5/SSP2, where the world will face intermediate adaptation and mitigation challenges, we see that most of the sub-national regions assessed would economically benefit from adaptation. We further see that the adaptation objectives differ in changes in relative risk and the level of adaptation that would take place. For instance, in the 'protection constant' adaptation objective we see that although the protection standards stay the same, the relative risk increases for most sub-national regions. This can be explained by the increase of the severity and frequency of the flood hazard due to sea-level rise and subsidence, and the increase of exposure of assets due to socioeconomic change. Compared to the 'optimize' adaptation objective, the 'protection constant' adaptation objective under-protects in most sub-national regions. In the 'absolute risk constant' adaptation objective we see that relative risk decreases in most sub-national regions while protection standards increase greatly. Due to climate change, socioeconomic change and subsidence, we see an increase in GDP exposed to flooding. Therefore, protection standards must increase vastly in order to meet the same level of absolute risk. In this adaptation objective, most sub-national regions are over-protected compared to the 'optimize' adaptation objective. In the 'relative risk constant' adaptation objective, we see that some sub-national regions are over-protected while other sub-national regions, for instance in Southeast Asia are under-protected. The 'optimize' adaptation objective shows the most economically feasible results in terms of maximizing NPV, and has the highest BCR in most regions. In the fossil-fuel based scenario of RCP8.5/SSP5, where adaptation will face high challenges and costs (van Vuuren et al., 2014), we see that higher protection standards are required in order to keep risk constant and to maximize NPV (see Supplementary Figures S3-6). The results of the adaptation objectives can be used as a first proxy to indicate in which sub-national regions adaptation through structural measures may be economically feasible. Moreover, the results indicate regions where adaptation is needed in order to maximize NPV and which objectives are under or over protecting sub-national regions compared to the 'optimize' adaptation objectives. Due to the scope of this study, local scale models and assessments should be used for the design and implementation of individual adaptation measures.'*

Technical corrections

19. List of typos/mistakes found:

- L. 26: 'compared to' stated twice

- L. 27: remove '.' after Raftery et al. (2017)

- L. 100: hydrologically

- L. 129: 30" x 30"

- L. 212: remove ',' after Jevrejeva et al. 2014

- L. 380: add ',' after South Asia

- L. 473: locations

20. The manuscript uses British English and American English interchangeably, one example being 'optimize', 'optimise', 'optimisation' etc in section 2.3.

- *Thanks -- all amended.*

---

## Author Comment (AC2) · 14 Feb 2020

**Response to referee 2**

**Manuscript for Natural Hazards and Earth System Sciences**

**Title: Global scale benefit-cost analysis of coastal flood adaptation to different flood risk drivers using structural measures**

**Authors**

Timothy Tiggeloven, Hans de Moel, Hessel C. Winsemius, Dirk Eilander, Gilles Erkens, Eskedar Gebremedhin, Andres Diaz Loaiza, Samantha Kuzma, Tianyi Luo, Charles Iceland, Arno Bouwman, Jolien van Huijstee, Willem Ligtvoet, Philip J. Ward

**General response**

We would like to thank referee 2 for the time taken to critically review our manuscript. We are very pleased that the referee finds the manuscript to be interesting and an incremental step forward. The referee raised a number of minor and modest corrections. The referee comments on the need to more explicitly state the innovative aspects and distinction of this study from previous assessments. We have addressed this concern along with detailed comments about the approach and methodology of this assessment. We believe that these revisions to the manuscript, and those detailed below, have led to a significant improvement in our manuscript. In the following pages, we respond to the comments of each referee point by point. Our responses are shown in italics.

**Referee: 2**

Overall this is an interesting paper. The approach undertaken is robust and I commend the authors for their nice study. The work builds on several previous assessments, and presents an incremental step forward, rather than a step change. However, I think it has some novel elements and is certainly worthy of publication in NHESS and results will be of interest to many. I have listed 5 modest corrections that I would like to see addressed and several minor ones.

- *Many thanks for the encouraging comments. We have addressed and clarified the modest and minor corrections in the manuscript, as described below.*

Modest corrections:

- In lines 50-55, you discuss the previous studies, and then go onto say what the objectives of your paper are. I think you need to make it clearer how your paper is distinct from these previous assessments. At the moment this does not come across strongly enough.

  *Thank you – we have addressed this in the revised manuscript by adding information about the distinction of this study. It now reads: 'Recent studies have shown that adaptation measures hold a large potential for significantly reducing this future flood risk (Diaz, 2016; Hinkel et al., 2014; Lincke and Hinkel, 2018). However, the number of global scale studies in which the benefits and costs of disaster risk reduction and adaptation are explicitly and spatially accounted for remains limited. Existing studies have assessed the effect of climate change, subsidence and/or socioeconomic change (Hallegatte et al., 2013; Hinkel et al., 2014; Nicholls et al., 2008; Vousdoukas et al., 2016), but have not included adaptation objectives or attributed flood risk drivers to adaptation costs. Lincke & Hinkel (2018) assessed the cost-effectiveness of structural*

protection measures against sea-level rise and population growth using the DIVA model. They found that structural adaptation measures are for 13% of the global coastline feasible to invest in. However, they did not include subsidence and attribution of drivers in their modelling scheme.

In this paper, we develop a model to evaluate the future benefits and costs of structural adaptation measures at the global scale. We use it to address the limitations of current studies addressed above, and thereby extend the current knowledge on the cost-effectiveness of structural adaptation measures in several ways. Firstly, we include subsidence due to groundwater extraction. Secondly, we assess the benefits and costs of several adaptation objectives. Thirdly, we attribute the costs of adaptation to different drivers (namely sea-level rise, subsidence and change in exposure)..'

- Please provide, on lines 94 to 99, more details of how exactly you have included the tropical cyclones. Over what period was this done? How did you covert along track data into spatially varying wind and pressure fields?

  *Thanks -- we have included the following information in section 2.1.1:* 'All tracks over the period 1979-2004 are used and converted into wind and pressure fields using the parametric Holland model (Delft3D-WES, 2019) in order to simulate aaalternative water levels using GTSM. These water levels are combined with the time series of GTSR by using the highest water level at each GTSM cell for each time step. Extreme values are estimated using a Gumbel extreme value distribution fit on the annual extremes.'

- Lines 124 to 133: I am not clear if these subsidence rates include glacial isostatic adjustment or not. Do they? Can you make this clear. I assume you are accounting for these effects. If not, then it significantly undervalues your results.

  *Thanks – in this study we only use groundwater extraction as a driver of subsidence, however, glacial isostatic adjustment is included in the regional sea-level approach. We have further clarified that in this study only groundwater extraction is used as a driver for subsidence.*

- Lines 172 – 185: I found the description of the protection standards confusing. Please can you improve this section. Has this approach me validated, in regions for example, where the protect standards are known exactly. How does these compare to what Hallegate et al (2013) used in coastal cites? You cite the Netherlands as havimg a value of 1000. What are the units? Years? Please add these.

  *Apologies for not referring to the validation of the coastal FLOPROS values, which can be found in the Supplementary Information. This is now amended. We have further clarified this section and moved information to the SI as suggested by referee 1. Indeed the units are in years and we have amended this.*

- Why is your analysis based on 2080, and not 2100? TO me, it would seem much more sensible to go to 2100?

*Thanks – the timeframe of the benefit-cost analysis is 2100 – we discount and accumulate all costs and benefits to this date. However, we have projections of climate and socioeconomic change at several points in time (2010, 2030, 2050 and 2080). Between these time-periods (and indeed after 2080) we extrapolate these values.*

Minor corrections:

Line25 – I would maybe update to the special IPCC report in 2019, which is a bit more up to date.

Line 27 – there is an extra full stop after the Raftery reference.

Line 29 – you could add 'and change in in tides.

*Thanks – all are amended.*